# Structure of *Mycobacterium tuberculosis* cytochrome *bcc* in complex with Q203 and TB47, two anti-TB drug candidates

Shan Zhou[1,2†], Weiwei Wang[3†], Xiaoting Zhou[3], Yuying Zhang[2], Yuezheng Lai[2], Yanting Tang[2], Jinxu Xu[2], Dongmei Li[1], Jianping Lin[1], Xiaolin Yang[3], Ting Ran[4], Hongming Chen[4], Luke W Guddat[5], Quan Wang[3], Yan Gao[3]*, Zihe Rao[1,2,3,6,7]*, Hongri Gong[2]*

[1]State Key Laboratory of Medicinal Chemical Biology, College of Pharmacy, Nankai University, Tianjin, China; [2]State Key Laboratory of Medicinal Chemical Biology, College of Life Sciences, Nankai University, Tianjin, China; [3]Shanghai Institute for Advanced Immunochemical Studies and School of Life Science and Technology, ShanghaiTech University, Shanghai, China; [4]Bioland Laboratory (Guangzhou Regenerative Medicine and Health - Guangdong Laboratory), Guangzhou, China; [5]School of Chemistry and Molecular Biosciences, The University of Queensland, Brisbane, Australia; [6]National Laboratory of Biomacromolecules, CAS Center for Excellence in Biomacromolecules, Institute of Biophysics, Beijing, China; [7]Laboratory of Structural Biology, Tsinghua University, Beijing, China

*For correspondence:
gaoyan@shanghaitech.edu.cn (YG);
raozh@mail.tsinghua.edu.cn (ZR);
gonghr@nankai.edu.cn (HG)

†These authors contributed equally to this work

Competing interest: The authors declare that no competing interests exist.

**Abstract** Pathogenic mycobacteria pose a sustained threat to global human health. Recently, cytochrome *bcc* complexes have gained interest as targets for antibiotic drug development. However, there is currently no structural information for the cytochrome *bcc* complex from these pathogenic mycobacteria. Here, we report the structures of *Mycobacterium tuberculosis* cytochrome *bcc* alone (2.68 Å resolution) and in complex with clinical drug candidates Q203 (2.67 Å resolution) and TB47 (2.93 Å resolution) determined by single-particle cryo-electron microscopy. *M. tuberculosis* cytochrome *bcc* forms a dimeric assembly with endogenous menaquinone/menaquinol bound at the quinone/quinol-binding pockets. We observe Q203 and TB47 bound at the quinol-binding site and stabilized by hydrogen bonds with the side chains of $_{QcrB}$Thr$^{313}$ and $_{QcrB}$Glu$^{314}$, residues that are conserved across pathogenic mycobacteria. These high-resolution images provide a basis for the design of new mycobacterial cytochrome *bcc* inhibitors that could be developed into broad-spectrum drugs to treat mycobacterial infections.

## Editor's evaluation

The authors describe the cryo EM structures of cytochrome bcc from mycobacteria in complex with clinical drug candidates Q203 and TB47, which are being trialled for the treatment of tuberculosis. The cryo EM structures were obtained by expressing and purifying a hybrid super complex; the bcc part from *M. tuberculosis* and the cytochrome oxidase part from *M. smegmatis*. The cryo EM structures show how the Q203 and TB47 inhibitors bind and block the binding of quinone at the Qo site, thus inhibiting electron-transfer. The complex structures should facilitate the structure-based development of Q203 and TB47 drug compounds against tuberculosis.

| | | Strain | Query cover (%) | Ident (%) |
|---|---|---|---|---|
| | | **Mycobacterium tuberculosis complex** | | |
| | | *Mycobacterium tuberculosis* | 100 | 100.00 |
| | | *Mycobacterium africanum* | 87 | 99.79 |
| | | *Mycobacterium bovis* | 99 | 87.46 |
| | | *Mycobacterium canettii* | 100 | 99.82 |
| | | *Mycobacterium microti* | 100 | 89.62 |
| | | *Mycobacterium orygis* | 100 | 99.82 |
| | | *Mycobacterium leprae* | 99 | 92.52 |
| | | *Mycobacterium marinum* | 99 | 88.71 |
| | | *Mycobacterium ulcerans* | 99 | 88.89 |
| | | **Mycobacterium avium complex** | | |
| | | *Mycobacterium avium* | 99 | 88.53 |
| | | *Mycobacterium intracellulare* | 99 | 86.92 |
| | | *Mycobacterium chimaera* | 85 | 82.09 |
| | | *Mycobacterium haemophilum* | 98 | 94.29 |
| | | *Mycobacterium xenopi* | 99 | 89.96 |
| | | *Mycobacterium kansasii* | 100 | 92.71 |
| | | *Mycobacterium simiae* | 100 | 91.99 |
| | | **Mycobacterium chelonae–abscessus complex** | | |
| | | *Mycobacterium abscessus subsp. abscessus* | 95 | 71.62 |
| | | *Mycobacterium abscessus subsp. bolletii* | 94 | 78.23 |
| | | *Mycobacterium chelonae* | 100 | 80.86 |
| | | *Mycobacterium fortuitum* | 98 | 98.00 |

Row labels (left margin): Slowly growing mycobacteria — Mycobacterium tuberculosis complex; Non-tuberculous mycobacteria; Rapidly growing mycobacteria.

Legend: ☐ True pathogens ☐ Opportunistic pathogens

**Figure 1.** Sequence similarity comparison of *M. tuberculosis* QcrB with other pathogenic mycobacteria.

## Introduction

Mycobacteria, which belong to the phylum Actinobacteria, have coevolved with humans over thousands of years (*Chisholm et al., 2016*). Approximately 200 species of mycobacteria have been identified that have diverse lifestyles, morphologies, and metabolic pathways (*Tortoli et al., 2017*). Mycobacteria can broadly be grouped into two categories: tuberculosis-causing mycobacteria and non-tuberculous mycobacteria (NTM). *Mycobacterium leprae* is often represented in a distinct genetic clade owing to its genetic and phenotypic differences compared to other mycobacterial species (*Cole et al., 2001*). Mycobacteria can be further classified into fast-growing and slow-growing species or species complexes; these assignments are according to the physiological, phenotypic and phylogenetic characteristics (*Rastogi et al., 2001*). Although nearly all mycobacteria are saprophytes or non-pathogenic to humans, a few species cause diseases, resulting in pulmonary and extra-pulmonary infections that can affect nearly all organs. Infections, which are caused by strict or opportunistic pathogenic mycobacteria (*Figure 1*), pose a sustained threat to human health. Of these, tuberculosis (TB), caused by *Mycobacterium tuberculosis* (*M. tuberculosis*), is the most serious, leading to ~1.2 million fatalities per year (*World Health Organization, 2019*). Infections involving other pathogenic mycobacteria, for example, *Mycobacterium abscessus* and *Mycobacterium avium complex*, are on the rise with some outnumbering those caused by *M. tuberculosis* in countries including the United States

(*Donohue, 2018*; *Johansen et al., 2020*). These infections are notoriously difficult to treat due to intrinsic or emerging resistance to many common antibiotics, thus exacerbating the challenge to find suitable drug targets.

Oxidative phosphorylation (OXPHOS) has gained interest as a target space for antibiotic drug development (*Cook et al., 2017*; *Hards et al., 2020*). In OXPHOS, the protein complexes of the electron transport chain (ETC) establish a proton motive force (PMF) across a biomembrane that drives the synthesis of adenosine triphosphate (ATP) by ATP synthase (*Mitchell, 1961*). Maintenance of PMF and ATP homeostasis is required for the survival of both replicative and non-replicative (often referred to as dormant) mycobacteria, and its dissipation leads to a rapid loss of cell viability and cell death (*Koul et al., 2008*; *Rao et al., 2008*). The reliance on the PMF and ATP homeostasis thus highlights the importance of the mycobacterial proton-pumping cytochrome $bcc$-$aa_3$ supercomplex, which consists of a $bcc$ menaquinol reductase (complex III, CIII) and an $aa_3$ oxidase (complex IV, CIV) that are tightly associated (*Gong et al., 2018*; *Kim et al., 2015*; *Megehee et al., 2006*). Several studies support the attractiveness of cytochrome $bcc$-$aa_3$ for mycobacterial drug development (*de Jager et al., 2020*; *Liu et al., 2019*; *Lu et al., 2019*; *Pethe et al., 2013*; *Scherr et al., 2018*). Given the strict sequence conservation of this complex (*Figure 1*), broad-spectrum activity of a therapeutic within the pathogenic mycobacteria is likely (*Lee et al., 2020b*). Interestingly, all the cytochrome $bcc$-$aa_3$ inhibitors published to date appear to target the QcrB subunit (*Figure 1*) of the cytochrome $bcc$ complex and are likely bound to the menaquinol-binding (Qp) site of the QcrB subunit (*Lee et al., 2020b*). The most advanced of these are Q203 and TB47, which have been shown to clear infections due to *M. tuberculosis* (*de Jager et al., 2020*; *Lu et al., 2019*; *Pethe et al., 2013*) and *Mycobacterium ulcerans* (*Liu et al., 2019*; *Scherr et al., 2018*). Q203 has recently completed phase II clinical trials for TB treatment (ClinicalTrials.gov number, NCT03563599; *de Jager et al., 2020*). TB47 has also been evaluated in a preclinical study (http://www.newtbdrugs.org/pipeline/clinical).

To progress an understanding of cytochrome $bcc$ and its interaction with new drug leads, here we have determined the atomic resolution cryo-electron microscopy (cryo-EM) structures of *M. tuberculosis* cytochrome $bcc$ alone (2.68 Å resolution) and in complex with Q203 (2.67 Å

**Table 1.** Cryo-electron microscopy data collection, refinement, and validation statistics of hybrid supercomplex.

| State | apo |
|---|---|
| *Data collection* | |
| Microscope | Titan Krios |
| Voltage (kV) | 300 |
| Magnification | 29,000× |
| Detector | Gatan K3 |
| Data collection software | SerialEM |
| Electron exposure (e⁻/Å²) | 60 |
| Defocus range (µm) | –1.2 to –1.8 |
| Pixel size (Å) | 0.82 |
| *Data processing* | |
| Number of micrographs | 4141 |
| Final particle images | 112,804 |
| Symmetry imposed | C1 |
| *Map resolution (Å)* | |
| Fourier shell correlation 0.143 threshold | 2.68 |
| *Refinement* | |
| Initial model used (PDB code) | 6ADQ |
| Map sharpening B factor (Å²) | –65.3 |
| d FSC model (0.143) masked | 2.5 |
| Map correlation coefficient | 0.89 |
| Mean CC for ligands | 0.78 |
| *Model composition* | |
| Non-hydrogen atoms | 42,279 |
| Protein residues | 5122 |
| | |
| | 9Y0: 2 |
| | CDL: 17 |
| | 9YF: 4 |
| | HEA: 4 |
| | HEC: 4 |
| | MQ9: 10 |
| | HEM: 4 |
| | PLM: 4 |
| | CU: 8 |
| | FES: 2 |
| *Ligands* | |
| *Root mean squared deviations* | |
| Bond lengths (Å) | 0.005 |
| Bond angles (°) | 1.057 |

*Table 1 continued on next page*

*Table 1 continued*

| Validation | |
|---|---|
| MolProbity score | 1.86 |
| Clashscore | 7.26 |
| Poor rotamers (%) | 0.05 |
| *Ramachandran plot* | |
| Favored (%) | 92.76 |
| Allowed (%) | 6.97 |
| Outliers (%) | 0.28 |
| Cβ outliers (%) | 0.00 |

resolution) and TB47 (2.93 Å resolution). These high-resolution structures will greatly accelerate efforts towards structure-guided drug discovery.

## Results and discussion
### Structure of *M. tuberculosis* cytochrome *bcc*

Considering that the hybrid supercomplex consisting of *M. tuberculosis* CIII and *Mycobacterium smegmatis* CIV can be stabilized as a functional assembly (***Kim et al., 2015***), we first chose to express and purify this complex to a high level of homogeneity (***Figure 2—figure supplement 1***). We confirmed this complex to be active with a turnover number of 23.3 ± 2.4 e⁻s⁻¹ (mean ± SD, n = 4), in agreement with the previous study that shows *M. tuberculosis* CIII can functionally complement native *M. smegmatis* CIII and maintain the growth of *M. smegmatis* (***Kim et al., 2015***). The structure of this hybrid supercomplex was determined by cryo-EM to an overall resolution of 2.68 Å, allowing that the components to be clearly assigned in the density map (***Figure 2—figure supplements 2 and 3***, ***Table 1***). The hybrid supercomplex including the *M. tuberculosis* CIII dimer forms a pseudo twofold symmetrical compact rod, but with a slight curvature in the membrane plane, as previously observed in the *M. smegmatis* CIII$_2$CIV$_2$ supercomplex (***Gong et al., 2018***; ***Wiseman et al., 2018***; ***Figure 2A and B***). The hybrid supercomplex is structurally similar to the supercomplex isolated from *M. smegmatis* (***Gong et al., 2018***; ***Wiseman et al., 2018***; ***Figure 2—figure supplement 4***), except that subunits LpqE and Unk (probably MSMEG_0987) (***Wiseman et al., 2018***) were not observed here. The absence of these two subunits may be due to their depletion during purification of the supercomplex or their map density signal was averaged to background noise during structural determination. As expected, the topology of *M. tuberculosis* cytochrome *bcc* and *M. smegmatis* CIV in the hybrid supercomplex is also similar to that of the equivalent subunits in the *M. smegmatis* CIII$_2$CIV$_2$ supercomplex (***Figure 2C and D***, ***Figure 2—figure supplements 4 and 5***). As such, there is no notable non-native contacts that resulted from the hybrid assembly in the hybrid supercomplex. This is attributable to the high structural similarity between the *M. tuberculosis* cytochrome *bcc* and *M. smegmatis* cytochrome *bcc*. In *M. tuberculosis* cytochrome *bcc*, QcrA has three transmembrane helices (TMHs) and has a 'U'-shaped structure (***Figures 2D and 3A***). The N-terminal region with TMH1/2 and the TMH3 make up the two arms of the 'U' structure. These arms are linked by the region located near the cytoplasmic side. Attached to $_{QcrA}$TMH3 is the C-terminal domain, which faces the periplasmic side of the membrane and holds the [2Fe-2S] cluster in place. QcrB has eight TMHs (***Figures 2D and 3B***). Four of these are responsible for burying two functionally important heme *b* cofactors (high potential heme $b_H$ and low potential heme $b_L$). Heme $b_L$ and heme $b_H$ are located between TMH I/II and TMH III/IV, respectively. Heme $b_L$ is near the periplasmic side and heme $b_H$ near the cytoplasmic side. QcrC is a transmembrane protein with a C-terminal TMH located between $_{QcrB}$TMH5 and $_{QcrB}$TMH7 (***Figures 2D and 3C***). The N-terminal periplasmic portion of QcrC can be divided into two heme-containing cytochrome *c* domains designated D1 and D2. The D1 domain protrudes out of the core of CIII, whereas the D2 domain interacts extensively with QcrA and QcrB. In the *M. smegmatis* CIII$_2$CIV$_2$ supercomplex, the cytochrome *cc* head domain of QcrC adopts an 'open' or a 'closed' conformation (***Gong et al., 2018***; ***Wiseman et al., 2018***). However, in this structure it is only the closed conformation (***Figure 2—figure supplement 4***). Considering the high topology similarity of the subunits and the arrangement of prosthetic groups compared to previous *M. smegmatis* CIII$_2$CIV$_2$ supercomplex, the mechanism of action of the hybrid supercomplex including *M. tuberculosis* cytochrome *bcc* is expected to be the same as that for the *M. smegmatis* CIII$_2$CIV$_2$ supercomplex (***Gong et al., 2018***).

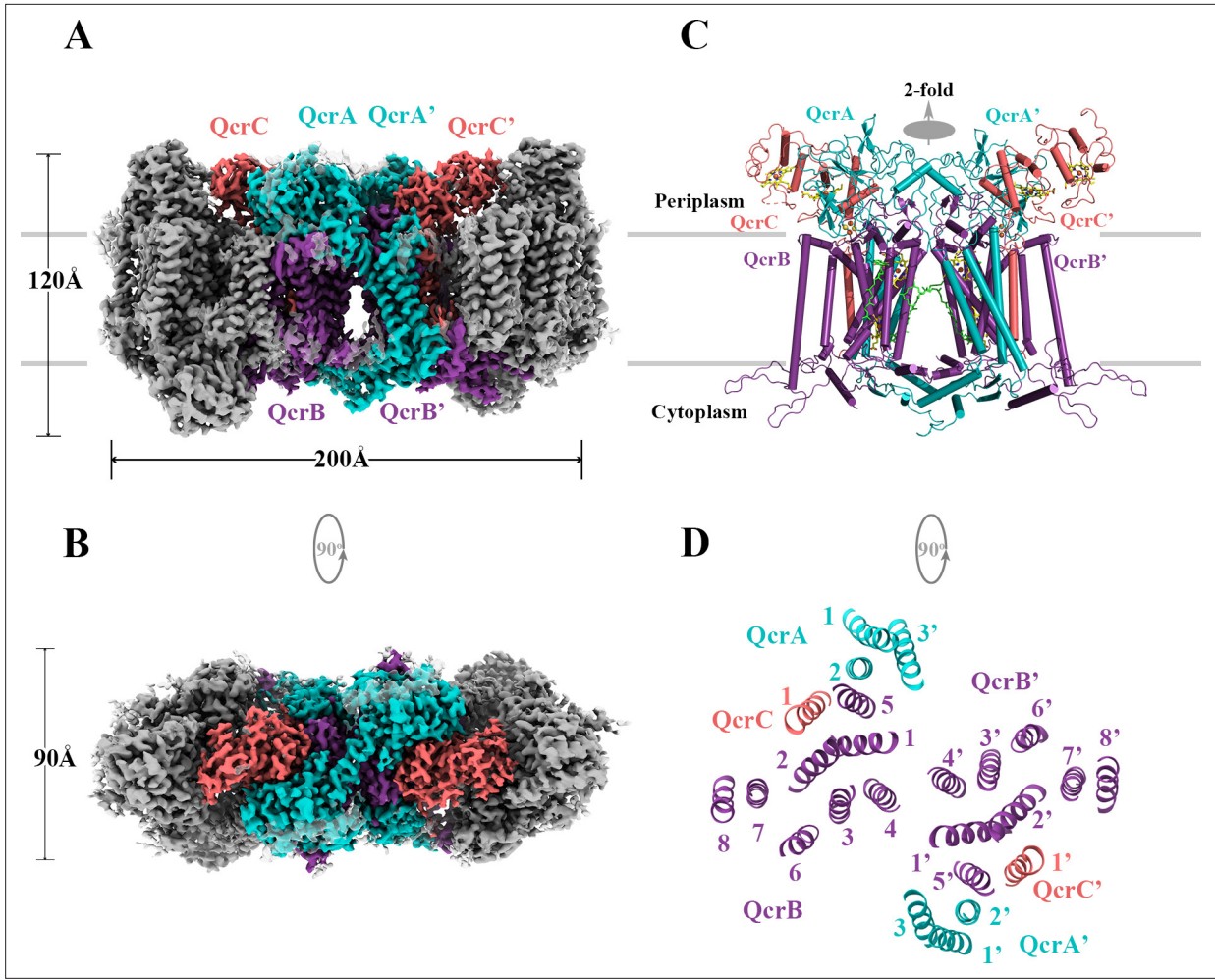

**Figure 2.** Overall architecture of the hybrid supercomplex. (**A**) Front view and (**B**) top view of the cryo-electron microscopy (cryo-EM) map of hybrid supercomplex at 2.68 Å resolution. QcrA, QcrB, and QcrC of the *M. tuberculosis* cytochrome *bcc* are colored teal, purple, and salmon, respectively. Other subunits of the hybrid supercomplex are in gray. (**C**) Cartoon representation of cytochrome *bcc*, using the same color scheme as above. The twofold symmetry of the dimer is depicted by the gray axis. The heme groups ($b_H$, $b_L$, $c_{D1}$, and $c_{D2}$) and menaquinone/menaquinol ($MK_P$/$MK_N$) are shown as stick models. The [2Fe-2S] clusters are shown as spheres. (**D**) A cross-sectional view (top) of the helices in the cytochrome *bcc* dimer.

The online version of this article includes the following source data and figure supplement(s) for figure 2:

**Source data 1.** Oxygen consumption of the hybrid supercomplex measures using Clark-type oxygen electrode.

**Figure supplement 1.** Purification and identification of the hybrid supercomplex consisting of *M. tuberculosis* CIII and *M. smegmatis* CIV.

**Figure supplement 1—source data 1.** The elution profile of the hybrid supercomplex from size-exclusion chromatography.

**Figure supplement 1—source data 2.** SDS-PAGE of the pooled fraction from the size-exclusion chromatography.

**Figure supplement 2.** Cryo-electron microscopy (cryo-EM) data processing of the apo hybrid supercomplex consisting of *M. tuberculosis* CIII and *M. smegmatis* CIV.

**Figure supplement 3.** Cryo-electron microscopy (cryo-EM) map quality assessment of the hybrid supercomplex.

**Figure supplement 4.** Structural comparison of the hybrid supercomplex III₂IV₂ and native CIII₂CIV₂ supercomplexes from *M. smegmatis*.

**Figure supplement 5.** Structural alignment between *M. tuberculosis* CIII and equivalent CIIIs from other species.

## Quinone and quinone-binding pockets of *M. tuberculosis* cytochrome *bcc*

Quinone-binding sites are essential to the function of respiratory chain complexes and thus are good targets for drug discovery (*Harikishore et al., 2021*; *Lee et al., 2020b*). Structurally diverse quinones such as ubiquinone and menaquinone (MK) bind in the mitochondrial ETC (*Zhang et al., 1998*) and mycobacterial ETC (*Gong et al., 2018*; *Wiseman et al., 2018*), respectively, indicating that there

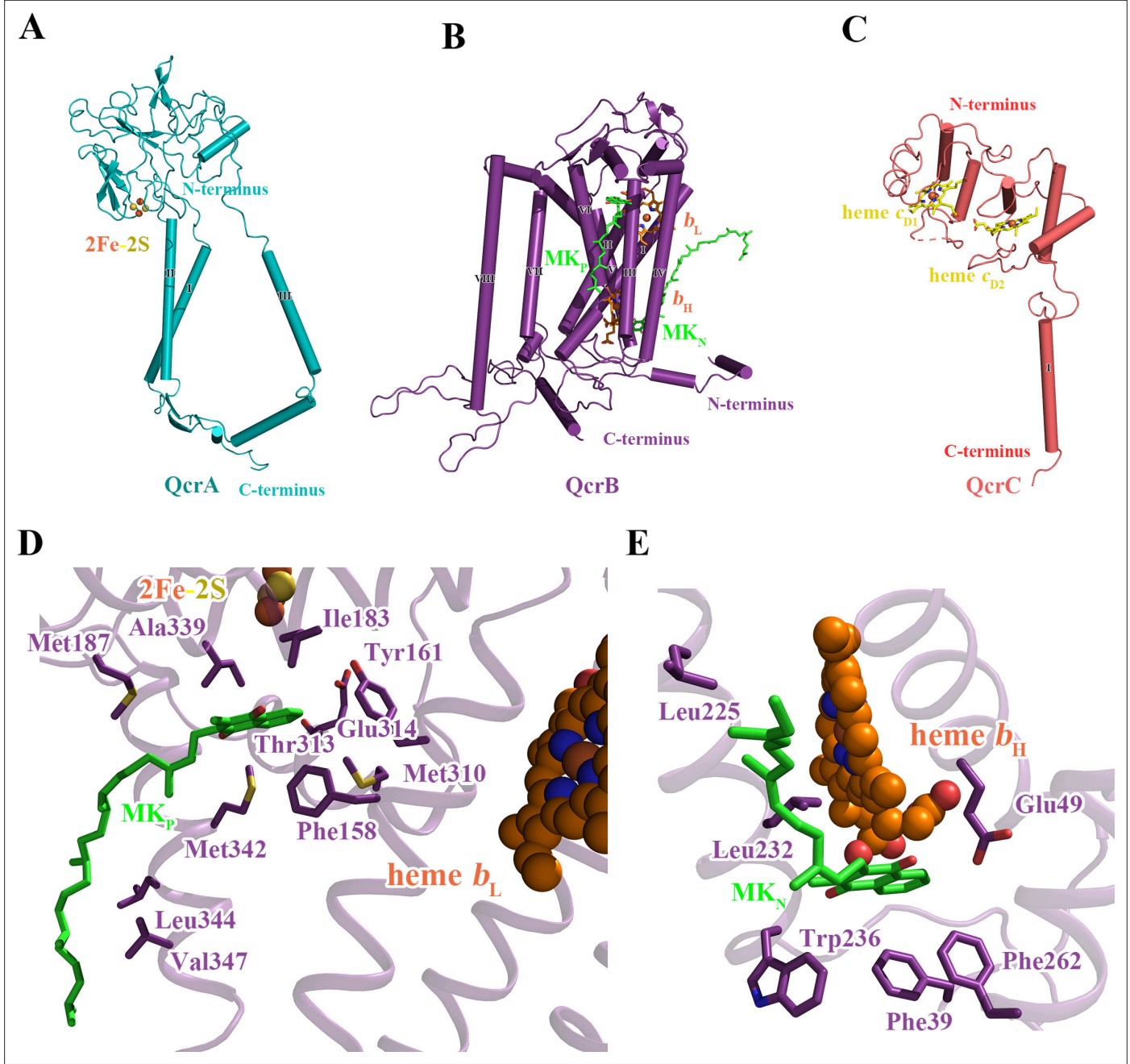

**Figure 3.** Structure of the *M. tuberculosis* cytochrome *bcc* subunits. Cartoon representation of the monomers of (**A**) QcrA, (**B**) QcrB, and (**C**) QcrC, with prosthetic groups. (**D**) The $Q_P$-binding site and (**E**) $Q_N$-binding site. The residues potentially involved in the binding of $MK/MKH_2$ are shown with side chains in stick model representation. $MK/MKH_2$ have their carbon atoms in green and are represented as stick models. The [2Fe-2S] and heme groups are shown as spheres and labeled.

The online version of this article includes the following figure supplement(s) for figure 3:

**Figure supplement 1.** Identification and comparison of $MK/MKH_2$ in the hybrid supercomplex.

**Figure supplement 2.** Comparison of *M. tuberculosis* CIII quinol-binding site (gray) with those from CIII from *S. cerevisiae* (PDB 3CX5) (blue) and (PDB 4PD4) (purple).

**Figure supplement 3.** Structural superposition of native (left) and docked (right) MK9 with the Q203 in the Qp site.

are structural differences in the quinone-binding sites of the different species. Structural differences in the ubiquinone-binding site are also observed when the mitochondrial and *Escherichia coli* respiratory chain complex IIs are compared (*Huang et al., 2021*). Therefore, it is feasible that species-specific quinone-binding-site inhibitors could be designed. In the present structure, two canonical quinone-binding sites are identified (*Figure 3D and E*): (i) the quinol oxidation site ($Q_P$ site) and (ii) the quinone reduction site ($Q_N$ site). In addition, other quinone-binding sites with quinone bound are also identified as observed in our previous study (*Gong et al., 2018*; *Figure 3—figure supplement 1*). These quinones are remote from the canonical quinone-binding sites and into the membrane space, suggesting that they have a structural rather than functional role. The $Q_P$ site responsible for menaquinol ($MKH_2$) oxidation is near heme $b_L$, whereas the $Q_N$ site responsible for MK reduction is close to heme $b_H$ (*Figure 3D and E*, *Figure 3—figure supplement 1*). The $Q_P$ site is at the center of an inverted triangle structure surrounded by helices (*Figure 3D*). One MK molecule was identified at this site with its naphthoquinone ring surrounded by hydrophobic residues, $_{QcrB}Phe^{158}$, $_{QcrB}Tyr^{161}$, $_{QcrB}Leu^{180}$, $_{QcrB}Ile^{183}$, $_{QcrB}Met^{310}$, and $_{QcrB}Met^{342}$. Its hydrophobic tail, which contains multiple isoprenoid groups, wraps around $_{QcrB}TMH6$ down to its cytoplasmic end and in so doing interacts with $_{QcrB}Met^{187}$, $_{QcrB}Ala^{339}$, $_{QcrB}Leu^{344}$, and $_{QcrB}Val^{347}$. The edge-to-edge distance from MK to heme $b_L$ is 15 Å, exceeding the 14 Å limit for efficient physiological electron transfer (*Page et al., 1999*). Furthermore, there are no observed hydrogen bonds to the carbonyl groups of MK, which are needed to help deprotonate $MKH_2$. In addition, structural superposition with the inhibitor-bound $bc_1$ complex also shows that the MK binds deeper into the $Q_P$ pocket in the $bc_1$ complex (*Birth et al., 2014*; *Solmaz and Hunte, 2008*) than in the currently described complex (*Figure 3—figure supplement 2*). Hence, the endogenous electron donor $MKH_2$ should bind deeper inside the pocket and close to polar residues such as $_{QcrB}Tyr^{161}$, $_{QcrB}Thr^{313}$, and $_{QcrB}Asp^{314}$ (*Figure 3D*) to facilitate electron transfer. In addition, MK could also be modeled to fit deeper inside the pocket (*Figure 3—figure supplement 3*). Thus, we speculate that what is observed here is the oxidized product as it leaves the $Q_P$ site. It is worth noting that all the reported inhibitors including Q203 (*Pethe et al., 2013*) and TB47 (*Lu et al., 2019*) are suggested to interact with this $Q_P$ site. In addition, the $Q_N$ site is formed largely

**Table 2.** Cryo-electron microscopy data collection, refinement, and validation statistics for the Q203-bound hybrid supercomplex.

| State | Q203 |
|---|---|
| *Data collection* | |
| Microscope | Titan Krios |
| Voltage (kV) | 300 |
| Magnification | 29,000× |
| Detector | Gatan K3 |
| Data collection software | SerialEM |
| Electron exposure (e⁻/Å²) | 60 |
| Defocus range (µm) | −1.2 to −1.8 |
| Pixel size (Å) | 0.82 |
| *Data processing* | |
| Number of micrographs | 3763 |
| Final particle images | 106,770 |
| Symmetry imposed | C1 |
| *Map resolution (Å)* Fourier shell correlation (FSC) 0.143 threshold | 2.67 |
| *Refinement* | |
| Initial model used (PDB code) | 6ADQ |
| Map sharpening B factor (Å²) d FSC model (0.143) masked | −70.0 2.6 |
| Map correlation coefficient | 0.88 |
| Mean CC for ligands | 0.76 |
| *Model composition* | |
| Non-hydrogen atoms | 42,695 |
| Protein residues | 5132 |
| | |
| | 9Y0: 2 |
| | CDL: 17 |
| | 9YF: 4 |
| | HEA: 4 |
| | HEC: 4 |
| | MQ9: 8 |
| | HEM: 4 |
| | PLM: 4 |
| | CU: 8 |
| | FES: 2 |
| | HUU (Q203): 2 |
| Ligands | |
| *Root mean squared deviations* | |

*Table 2 continued on next page*

*Table 2 continued*

| | |
|---|---|
| Bond lengths (Å) | 0.003 |
| Bond angles (°) | 0.659 |
| *Validation* | |
| MolProbity score | 1.84 |
| Clashscore | 7.97 |
| Poor rotamers (%) | 6.23 |
| *Ramachandran plot* | |
| Favored (%) | 93.07 |
| Allowed (%) | 6.61 |
| Outliers (%) | 0.31 |
| Cβ outliers (%) | 0.00 |

by $_{QcrB}$TMH1, $_{QcrB}$TMH4, $_{QcrB}$TMH5 and one loop region of QcrB (*Figure 3E*). The head group of MK is bound in this pocket interacting with $_{QcrB}$Phe$^{39}$, $_{QcrB}$Glu$^{49}$, $_{QcrB}$Leu$^{225}$, $_{QcrB}$Leu$^{232}$, $_{QcrB}$Trp$^{236}$, and $_{QcrB}$Phe$^{262}$, and its long hydrophobic tail extends along $_{QcrB}$TMH1 towards the periplasmic side. MK/MKH$_2$ are part of the Q-cycle hypothesis and essential for electron transfer in the cytochrome *bcc* complex (*Gong et al., 2018*). Given the crossspecies activity of this complex (*Lee et al., 2020b*) and high homology of the QcrB subunits across mycobacterial pathogens (*Figure 1*), these data open the way for the discovery of broad-spectrum mycobacterial agents based on rational, structure-based inhibitor design principles.

## Q203 interactions in *M. tuberculosis* cytochrome *bcc* binding pocket

Q203 has recently been subjected to a phase II clinical study for *M. tuberculosis* treatment (*de Jager et al., 2020*). This compound has also been shown to be strongly bactericidal against *M. ulcerans* (*Scherr et al., 2018*). It is suggested to be an inhibitor that competes with endogenous substrate binding ($Q_P$ site) of the cytochrome *bcc* complex (*Pethe et al., 2013*), but this hypothesis is yet to be verified by direct experimental evidence. To obtain atomic information on the mode of binding of Q203 to cytochrome *bcc*, we have determined the structure of the hybrid supercomplex as described above in the presence of Q203 by cryo-EM to an overall resolution of 2.67 Å (*Figure 4—figure supplements 1 and 2*, *Table 2*). We observe that close to the Qp-binding pocket within the membrane of each QcrB of cytochrome *bcc* there is density for Q203 (*Figure 4A and B*). All of the Q203 molecules fill each QcrB subunit binding deeply into the $Q_P$ pocket and with identical binding modes. The key interactions that anchor Q203 are (i) a hydrogen bond between the hydroxyl oxygen of the side chain of $_{QcrB}$Thr$^{313}$ and the amine in the carboxamide linker of Q203 (3.0 Å), (ii) a halogen bond between the chlorine atom of the heterocyclic group and an ordered water molecule that simultaneously forms a hydrogen bond with the hydroxyl oxygen of the side chain of $_{QcrB}$Tyr$^{164}$ (*Figure 4—figure supplement 3*), (iii) a hydrogen bond between the side chain of $_{QcrB}$Glu$^{314}$ and the nitrogen atom in the imidazopyridine ring (3.0 Å), and (iv) a hydrogen bond between the side chain of $_{QcrA}$His$^{375}$ and the nitrogen atom in the imidazopyridine ring (2.98 Å) (*Figure 4C*). In addition, the carbon atoms of Q203 make hydrophobic interactions with $_{QcrB}$Gly$^{175}$, $_{QcrB}$Ala$^{179}$, $_{QcrB}$Leu$^{180}$, $_{QcrB}$Thr$^{184}$, $_{QcrB}$Ser$^{304}$, $_{QcrB}$Pro$^{306}$, $_{QcrB}$Met$^{310}$, $_{QcrB}$Ala$^{317}$, and $_{QcrB}$Met$^{342}$. These extensive interactions are in agreement with the fact that the activity of the supercomplex is inhibited by Q203 in vitro according to the DMNQH$_2$/oxygen oxidoreductase activity assay. After addition of Q203, the turnover number of the hybrid supercomplex reduces to 5.8 ± 2.4 e$^-$s$^{-1}$ from 23.3 ± 2.4 e$^-$s$^{-1}$ (mean ± SD, n = 4; *Figure 4—figure supplement 4*). In addition, functional studies have shown that substitution of $_{QcrB}$Thr$^{313}$ to alanine confers Q203 resistance (*Pethe et al., 2013*). The binding of Q203 involves residues from both QcrA and QcrB. Due to the need to form stabilizing interactions between subunits, resistance may be more difficult to achieve here than if the binding site is within only one subunit. Consistently, the mapping of reported mutations in Q203-resistant *M. tuberculosis* reveals that they are positioned directly where Q203 binds in this structure (*Lupien et al., 2020*; *Figure 4—figure supplement 5*).

## TB47-binding mode of *M. tuberculosis* cytochrome *bcc*

TB47, also currently being evaluated in preclinical studies, has been suggested to target the QcrB of cytochromes *bcc* from *M. tuberculosis* (*Lu et al., 2019*) and *M. ulcerans* (*Liu et al., 2019*). The 2.93 Å cryo-EM map shows density for TB47 and confirms that it binds in the same location as Q203 (*Figure 5A and B*, *Figure 5—figure supplements 1 and 2*, *Table 3*). Three hydrogen bond interactions are observed involving the side chains of $_{QcrB}$Thr$^{313}$, $_{QcrB}$Glu$^{314}$, and $_{QcrA}$His$^{375}$. Similar interactions are also observed when Q203 binds (*Figure 5C*). $_{QcrB}$Tyr$^{161}$, $_{QcrB}$Leu$^{171}$, $_{QcrB}$Gly$^{175}$, $_{QcrB}$Ala$^{179}$, $_{QcrB}$Leu$^{180}$,

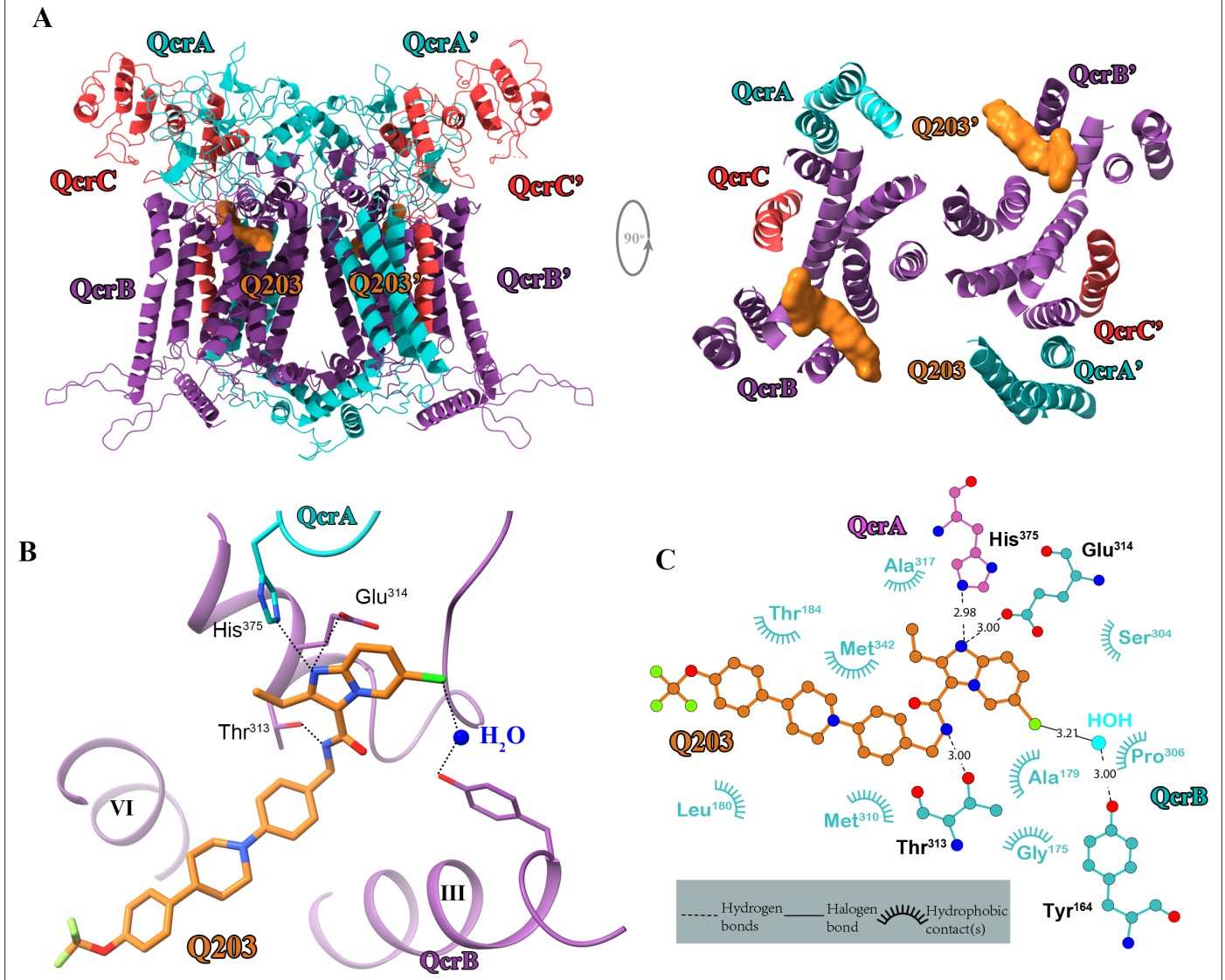

**Figure 4.** Cryo-electron microscopy (cryo-EM) structure of the hybrid supercomplex in the presence of Q203. (**A**) Side (left) and top (right) views of the cryo-EM structure of the *M. tuberculosis* cytochrome *bcc* complex presented as a cartoon representation. Q203 (orange) is bound to the Qp site. (**B**) Visualization of densities for Q203. Hydrogen bonds are shown as dotted lines. (**C**) Plot of distances of various parts of Q203 to residues in the Qp site as determined using LIGPLOT (https://www.ebi.ac.uk/thornton-srv/software/LIGPLOT/).

The online version of this article includes the following source data and figure supplement(s) for figure 4:

**Figure supplement 1.** Cryo-electron microscopy (cryo-EM) data processing of the hybrid supercomplex consisting of *M. tuberculosis* CIII and *M. smegmatis* CIV in the presence of Q203.

**Figure supplement 2.** Cryo-electron microscopy (cryo-EM) map quality assessment of *M. tuberculosis* cytochrome *bcc* complex.

**Figure supplement 3.** The densities for Q203, $H_2O$, and $_{QcrB}Tyr^{164}$.

**Figure supplement 4.** Rate of $O_2$ reduction by the hybrid supercomplex before and after addition of Q203.

**Figure supplement 4—source data 1.** Oxygen consumption of the hybrid supercomplex after addition of Q203 measures using Clark-type oxygen electrode.

**Figure supplement 5.** Reported mutations in Q203-resistant *M. tuberculosis*.

$_{QcrB}Thr^{184}$, $_{QcrB}Met^{187}$, $_{QcrB}Leu^{194}$, $_{QcrB}Ser^{304}$, $_{QcrB}Gly^{305}$, and $_{QcrB}Met^{342}$ also contribute to TB47 binding, largely through hydrophobic interactions (***Figure 5C***). Unlike when Q203 binds (***Figure 4—figure supplement 3***), there is no interaction between $_{QcrB}Tyr^{164}$ and TB47 (***Figure 5—figure supplement 3***). However, after addition of TB47, the values of turnover number for the hybrid supercomplex reduced

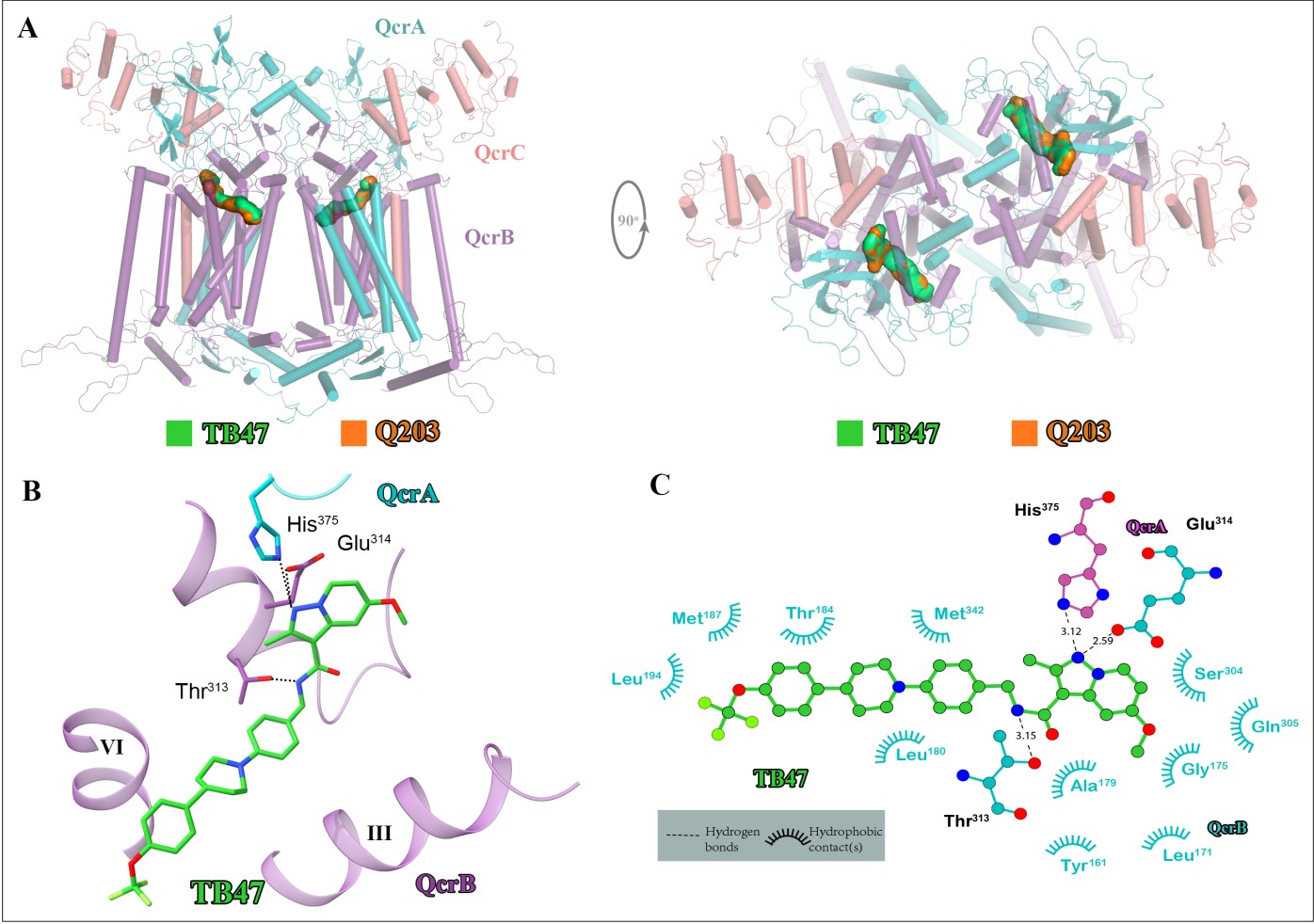

**Figure 5.** Cryo-electron microscopy (cryo-EM) structure of the hybrid supercomplex in the presence of TB47. (**A**) Side (left) and top (right) views of the cryo-EM structure of the *M. tuberculosis* cytochrome *bcc* complex presented as a cartoon representation. TB47 (green) and Q203 (orange) are bound to the Qp site. (**B**) Visualization of the density for TB47. Hydrogen bonds are shown as dotted lines. (**C**) Plot of distances of various parts of TB47 to residues in the Qp site were determined using LIGPLOT (https://www.ebi.ac.uk/thornton-srv/software/LIGPLOT/).

The online version of this article includes the following source data and figure supplement(s) for figure 5:

**Figure supplement 1.** Cryo-electron microscopy (cryo-EM) data processing of the hybrid supercomplex consisting of *M. tuberculosis* CIII and *M. smegmatis* CIV in the presence of TB47.

**Figure supplement 2.** Cryo-electron microscopy (cryo-EM) map quality assessment for the hybrid *M. tuberculosis* cytochrome *bcc* complex.

**Figure supplement 3.** The densities for TB47 and ₍QcrB₎Tyr¹⁶⁴.

**Figure supplement 4.** Rate of $O_2$ reduction by the hybrid supercomplex before and after addition of TB47.

**Figure supplement 4—source data 1.** Oxygen consumption of the hybrid supercomplex after addition of TB47 measures using Clark-type oxygen electrode.

**Figure supplement 5.** Reported mutations in TB47-resistant *M. tuberculosis*.

to 5.1 ± 2.9 e⁻s⁻¹ (mean ± SD, n = 4; *Figure 5—figure supplement 4*), a similar value that was observed when Q203 binds. Thus, the absence of this does not greatly diminish inhibition. A mutation in TB47-resistant *M. smegmatis* (*M. tuberculosis*: H195Y) is close to the Qp-binding site (*Lu et al., 2019*; *Figure 5—figure supplement 5*). As a result of the change in shape, it would appear to contribute to steric interference with the binding of TB47, thus accounting for the observed resistance.

**Table 3.** Cryo-electron microscopy data collection, refinement, and validation statistics for the TB47-bound hybrid supercomplex.

| State | TB47 |
|---|---|
| *Data collection* | |
| Microscope | Titan Krios |
| Voltage (kV) | 300 |
| Magnification | 29,000× |
| Detector | Gatan K3 |
| Data collection software | SerialEM |
| Electron exposure (e⁻/Å²) | 60 |
| Defocus range (µm) | –1.2 to –1.8 |
| Pixel size (Å) | 0.82 |
| *Data processing* | |
| Number of micrographs | 2698 |
| Final particle images | 169,988 |
| Symmetry imposed | C1 |
| *Map resolution (Å)* Fourier shell correlation (FSC) 0.143 threshold | 2.93 |
| *Refinement* | |
| Initial model used (PDB code) | 6ADQ |
| Map sharpening B factor (Å²) d FSC model (0.143) masked | –97.5 2.9 |
| Map correlation coefficient | 0.90 |
| Mean CC for ligands | 0.79 |
| *Model composition* | |
| Non-hydrogen atoms | 42,679 |
| Protein residues | 5119 |
| | 9Y0: 2 |
| | CDL: 17 |
| | 9YF: 4 |
| | HEA: 4 |
| | HEC: 4 |
| | MQ9: 8 |
| | HEM: 4 |
| | PLM: 4 |
| | CU: 8 |
| | FES: 2 |
| | HV0 (TB47): 2 |
| Ligands | |
| *Root mean squared deviations* | |
| Bond lengths (Å) | 0.005 |

*Table 3 continued on next page*

*Table 3 continued*

| | |
|---|---|
| Bond angles (°) | 0.739 |
| *Validation* | |
| MolProbity score | 1.87 |
| Clashscore | 8.75 |
| Poor rotamers (%) | 6.23 |
| *Ramachandran plot* | |
| Favored (%) | 92.39 |
| Allowed (%) | 7.25 |
| Outliers (%) | 0.36 |
| Cβ outliers (%) | 0.00 |

## Specificity of Q203 and TB47 for mycobacterial cytochrome *bcc* complex

The basis for the high specificity of Q203 and TB47 toward the Qp site of mycobacterial cytochromes *bcc* becomes apparent in the structural comparison between the QcrB subunit of *M. tuberculosis* and counterparts from other species (***Figure 6***). The highly conserved residues that are involved in the binding of these two molecules in this region (***Figure 6—figure supplement 1***) suggest a consistent overall fold and binding site exists in mycobacteria. This is also in agreement with the fact that Q203 and TB47 show antimycobacterial activity across many species (***de Jager et al., 2020***; ***Liu et al., 2019***; ***Lu et al., 2019***; ***Pethe et al., 2013***; ***Scherr et al., 2018***). It is worth noting that the structures of cytochrome *bcc* from *M. tuberculosis* and *M. smegmatis* have high similarity (***Figure 2—figure supplement 5***), and no steric hindrance is observed between the Q203 and *M. smegmatis* cytochrome *bcc* (***Figure 6***). This observation indicates that Q203 should have a similar binding mechanism and a similar effect on the activity of cytochrome *bcc* from *M. smegmatis* and *M. tuberculosis*. This is in good agreement with previous antimycobacterial activity data and inhibition data for the *bcc* complexes from *M. smegmatis* and *M. tuberculosis* (***Gong et al., 2018***; ***Lu et al., 2018***). In contrast, in other prokaryotic, eukaryotic and human Qp-binding pockets, for example, from *Saccharomyces cerevisiae* (***Lange and Hunte, 2002***), *Rhodobacter sphaeroides* (***Esser et al., 2008***), and human (***Guo et al., 2017***), many of the observed interactions would not be possible (***Figure 6***). This suggests that Q203 and TB47 should have low-binding affinity toward its counterpart QcrB in

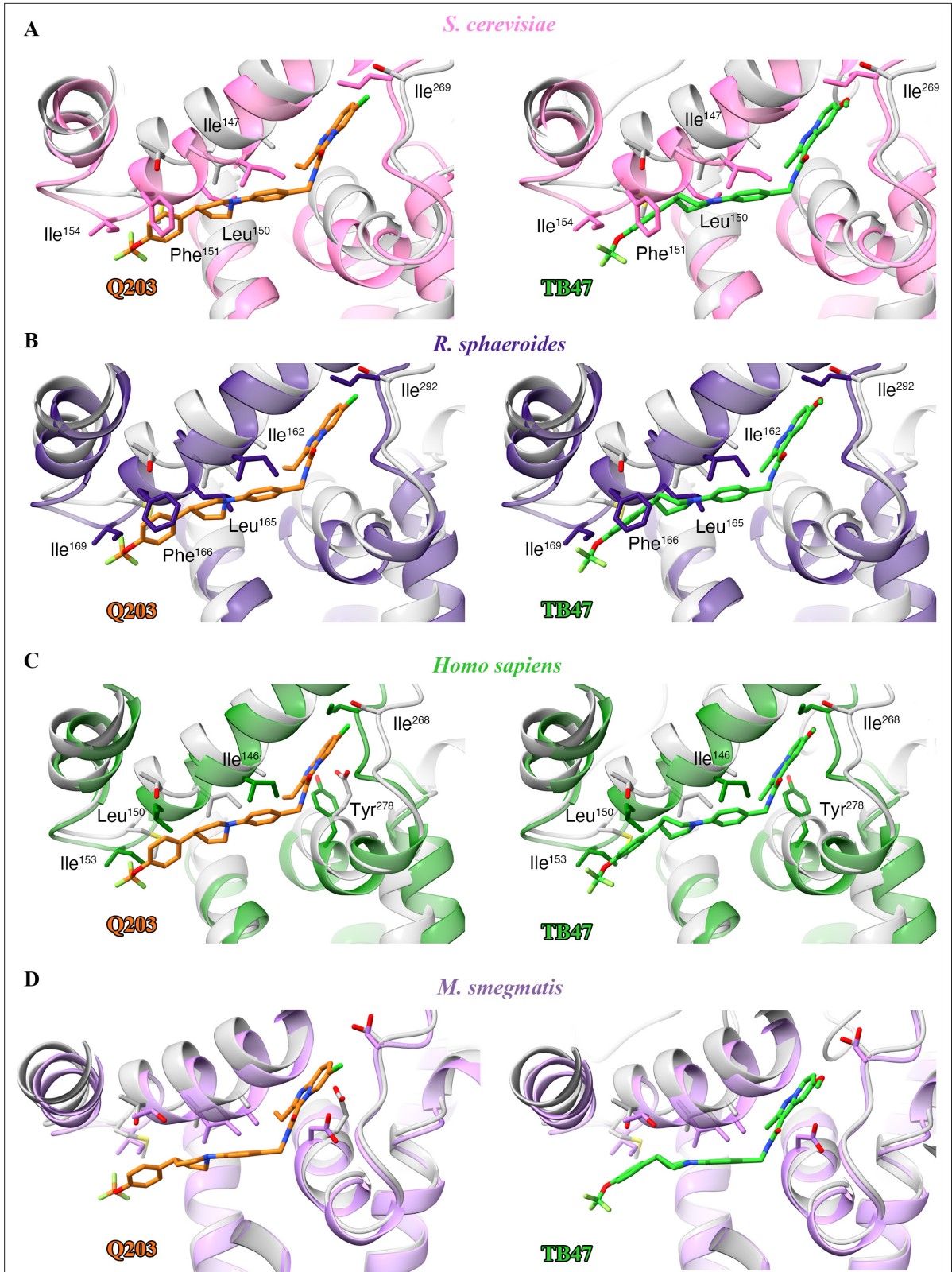

**Figure 6.** Structural alignment between the *M. tuberculosis* Qp-binding pocket where Q203 or TB47 binds with homologous subunits from four other species. These subunits are from (**A**) *S. cerevisiae* (pink, PDB: 1KYO), (**B**) *R. sphaeroides* (blue, PDB: 2QJP), (**C**) *Homo sapiens* (green; PDB: 5XTE), and (**D**) *M. smegmatis* (violet, PDB: 6ADQ). Residues causing steric clashes in the homologous subunits are labeled.

*Figure 6 continued on next page*

*Figure 6 continued*

The online version of this article includes the following source data and figure supplement(s) for figure 6:

**Figure supplement 1.** Sequence alignment of *M. tuberculosis* QcrB with their counterparts in other species including *Homo sapiens*.

**Figure supplement 2.** Molecular dynamics simulation plot for the root mean squared deviation (RMSD) of the heavy atoms in the inhibitor (**A**) and main chain atoms of QcrB (**B**).

**Figure supplement 2—source data 1.** Root mean squared deviation of the inhibitor and protein of Q203-bound QcrB complex.

non-mycobacterial bacteria and in eukaryotes. Even if there is some flexibility in the Qp-binding pocket that enables some level of binding, key residues that enable the binding of Q203 and TB47 in the mycobacteria are not present in other bacteria and eukaryotes (*Figure 6—figure supplement 1*). In addition, the binding free energies of Q203 between wild-type and mutant hybrid supercomplex were calculated and used to further elucidate the importance of some key residues that could play a role in determining the inhibitor specificity. According to a previous functional study (*Pethe et al., 2013*), structural comparison (*Figure 6*), and sequence alignment (*Figure 6—figure supplement 1*), $_{QcrB}$Thr$^{313}$ and $_{QcrB}$Glu$^{314}$ of hybrid supercomplex were selected and mutated to their counterparts in the human supercomplex, namely residues Ala and Tyr, respectively. It is believed that Ala would decrease the affinity of Q203 (*Figure 4*; *Pethe et al., 2013*) and the Tyr could lead to steric hindrance of Q203 binding (*Figure 6*). 75 ns simulations were used for the calculation of binding free energy (*Figure 6— figure supplement 2*). The different values for the root mean squared deviation (RMSD) suggest that three mutants might have the different effects on the structural conformation of QcrB, and the dual mutant (Thr313Ala and Glu314Tyr) induced significantly structural changes compared to the wild-type complex. The 75 ns simulation was further equally divided into three phases, and the average values were used as the final relative binding free energy (*Table 4*), which was calculated as the binding free energy of Q203 in the mutant minus that of the wild-type systems. It is suggested that the wild-type system has stronger affinity than the mutant systems (Thr313Ala or Glu314Tyr). The relative free energy of binding calculation for Q203 in the dual mutant (Thr313Ala and Glu314Tyr) indicates that its binding affinity is significantly weaker compared to the wild-type or single-mutant systems. Therefore, the suggestions drawn from the calculation of binding free energy of Q203 provide further evidence for specificity of Q203 inhibition. In summary, these observations correlate with the observed low general antibacterial activity and low cytotoxicity of Q203 and TB47 (*Liu et al., 2019*; *Lu et al., 2019*; *Pethe et al., 2013*; *Scherr et al., 2018*).

## Implications of the Q203 and TB47 inhibitory mechanism

To gain further insights into the mechanism of action of Q203, we compared the structures of *M. tuberculosis* cytochrome *bcc* in the presence and absence of Q203 (*Figure 7—figure supplement 1A*). The structure of apo cytochrome *bcc* is almost identical with the Q203-bound structure (rmsd of 0.50 Å for all Ca atoms), which suggests that Q203 binding does not significantly affect the overall architecture of cytochrome *bcc*. A comparison of the Q203-bound and Q203-free cytochrome *bcc* structures shows residues involved in the binding pocket move outward, thus adapting to the shape of Q203 (*Figure 7—figure supplement 1B*). Specifically, the side chains of $_{QcrB}$Ser$^{304}$, $_{QcrB}$Glu$^{313}$, $_{QcrB}$Glu$^{314}$, and $_{QcrB}$Met$^{342}$ undergo significant conformational changes to form hydrogen bonds with Q203. The binding of TB47 to *M. tuberculosis* cytochrome *bcc* also induces very similar conformational changes in the Qp-binding pocket to those seen for Q203 (rmsd of 0.454 Å for all Ca atoms) (*Figure 7—figure supplement 1B*). Differences in binding are due to the different ethyl group and methyl moieties in the head groups of Q203 and TB47. It is also important to note that one endogenous substrate

**Table 4.** Relative binding free energy (kcal/mol) for Q203 in three mutants of QcrB compared to the wild-type (WT).

| Mutant | 25–50 ns | 50–75 ns | 75–100 ns | Average | Standard deviation |
|---|---|---|---|---|---|
| T313A | 6.32 | 11.01 | 8.11 | 8.48 | 2.37 |
| T314Y | 4.54 | 6.00 | 8.07 | 6.20 | 1.77 |
| T313A + E314Y | 10.28 | 16.30 | 13.64 | 13.41 | 3.02 |

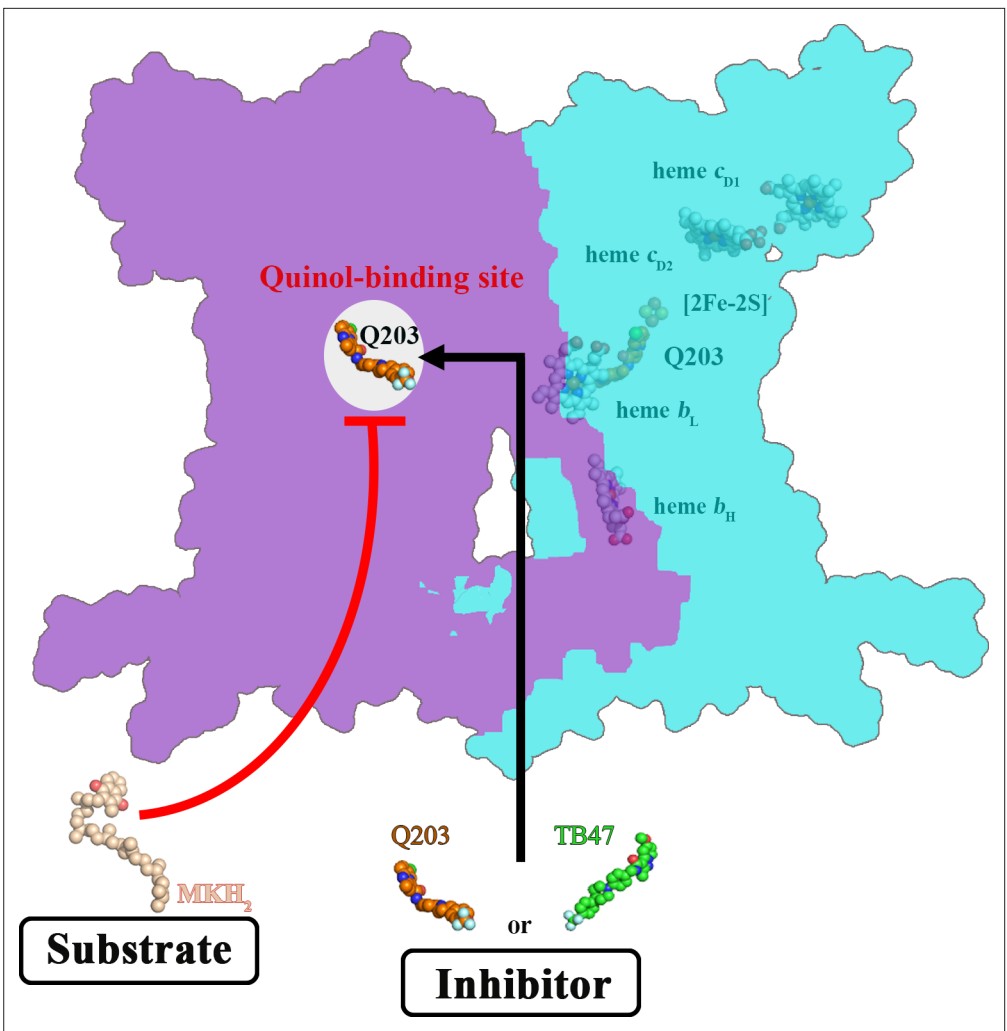

**Figure 7.** Schematic of *M. tuberculosis* cytochrome *bcc* inhibition by Q203 and TB47. The two monomers of *M. tuberculosis* cytochrome *bcc* are colored magenta and cyan, respectively. The binding of Q203 (orange spheres) or TB47 (green spheres) prevents substrate access (gray spheres).

The online version of this article includes the following figure supplement(s) for figure 7:

**Figure supplement 1.** Comparison of apo and Q203/TB47-bound structures of *M. tuberculosis* cytochromes *bcc*.

**Figure supplement 2.** Sequence alignment for the $Q_P$ site within pathogenic mycobacteria.

molecule is also bound to the Qp site in the apo structure of cytochrome *bcc*, which potentially affects the evaluation of the conformational changes upon the binding of Q203 or TB47.

When analyzing the superimposed structures (*Figure 7—figure supplement 1*), it is apparent that Q203 and TB47 act competitively with the quinol binding as they almost completely occupy the Qp pocket. We therefore conclude that Q203 and TB47 are bona fide analogs of the substrate, and thus ultimately function by hindering the downstream synthesis of ATP (*Figure 7*). These two compounds are also highly bactericidal against *M. ulcerans*, almost certainly targeting the Qp-binding site (*Liu et al., 2019*; *Scherr et al., 2018*). In summary, the sequences of the QcrB subunits have high homology across pathogenic mycobacteria (*Lee et al., 2020b*) and the essential residues ($_{QcrB}$Glu$^{313}$ and $_{QcrB}$Glu$^{314}$) that are involved in hydrogen-bonding interactions with the inhibitors (*Pethe et al., 2013*; *Scherr et al., 2018*) are conserved across pathogenic mycobacteria (*Figure 7—figure supplement 2*).

## Conclusions

We have determined the apo- and Q203 and Tb47-bound structures of a hybrid pathogenic *M. tuberculosis/M. smegmatis* cytochrome *bcc* complex. The study shows the structural features of *M. tuberculosis* cytochrome *bcc* and how it is specifically inhibited by Q203 and TB47. The extensive interactions between Q203 or TB47 and the Qp-binding pocket account for the highly specific binding of these two inhibitors to pathogenic *M. tuberculosis* cytochrome *bcc* compared to eukaryotic counterparts. Two conservative residues involved with the formation of hydrogen bonds are observed across the pathogenic mycobacteria. These structures provide a long-sought basis for rational, structure-based inhibitor design to accelerate the development of Q203 and TB47 analogs as drug leads for myco-bacterial infections.

# Materials and methods

**Key resources table**

| Reagent type (species) or resource | Designation | Source or reference | Identifiers | Additional information |
|---|---|---|---|---|
| Gene (*Mycobacterium tuberculosis*) | qcrC | Mycobrowser | Rv2194 | https://mycobrowser.epfl.ch/genes/Rv2194 |
| Gene (*M. tuberculosis*) | qcrA | Mycobrowser | Rv2195 | https://mycobrowser.epfl.ch/genes/Rv2195 |
| Gene (*M. tuberculosis*) | qcrB | Mycobrowser | Rv2196 | https://mycobrowser.epfl.ch/genes/Rv2196 |
| Strain, strain background (*Mycobacterium smegmatis*) | mc$^2$ 51 | *Li et al., 2014* | | |
| Genetic reagent (include species here) | pVV16-QcrCAB-His10 | This paper | | Construct contains the qcrCAB operon encoding three subunits |
| Commercial assay or kit | ClonExpress II One Step Cloning Kit | Vazyme | C112-01 | |
| Chemical compound, drug | Q203 | MCE | HY-101040 | Prepare stock solution in DMSO |
| Chemical compound, drug | TB47 | *Lu et al., 2019* | | Prepare stock solution in DMSO |
| Chemical compound, drug | LMNG | Anatrace | NG310 | |
| Chemical compound, drug | Digitonin | BIOSYNTH | D-3200 | |
| Software, algorithm | SerialEM | *Mastronarde, 2003* | | Version 3.6 |
| Software, algorithm | MotionCor2 | *Zheng et al., 2017* | | Version 1.2.1 |
| Software, algorithm | RELION | *Zivanov et al., 2019* | | Version 3.03 |
| Software, algorithm | cryoSPARC | *Punjani et al., 2017* | | Version 3.2.0 |
| Software, algorithm | Phyre2 | *Kelley et al., 2015* | | Version 2.0 |
| Software, algorithm | UCSF Chimera | *Pettersen et al., 2004* | | Version 1.12 |
| Software, algorithm | COOT | *Emsley et al., 2010* | | Version 0.8.9 |
| Software, algorithm | PHENIX | *Adams et al., 2010* | | Version 1.16 |
| Software, algorithm | PyMOL | *Schrödinger LLC, 2017* | | Version 2.0 |
| Software, algorithm | ChemDraw | *Li et al., 2004* | | Version 19.0 |

## Expression of hybrid supercomplex consisting of *M. tuberculosis* CIII and *M. smegmatis* CIV

The hybrid supercomplex was obtained according to a previous study (*Kim et al., 2015*) but with some modifications. The *M. tuberculosis* cytochrome *bcc* complex is encoded by three putative genes (Rv2194-2196). Genes were amplified from H37Rv genomic DNA by PCR using Phanta Max DNA polymerase (Vazyme), and two-step PCR was used to inset a 10× His tag at the C-terminus of the

qcrB (Rv2196). Genes encoding the entire cytochrome *bcc* complex operon were then cloned into the pVV16 expression vector. The resultant plasmid was transformed into *M. smegmatis* mc[2] 51 (*Li et al., 2014*) cells whose qcrCAB operon encoding *M. smegmatis* cytochrome *bcc* had already been knocked out. The cells were cultivated in Luria-Bertani broth (LB) liquid media supplemented with 50 μg/mL hygromycin, 25 μg/mL streptomycin, and 0.1% Tween 80. Cell pellets were harvested by centrifugation when the cells were grown to an optical density (OD$_{600}$ of 1.0–1.2) at 37°C (220 rpm). Harvested cells were frozen at –80°C until use.

## Purification of the hybrid supercomplex

Cell pellets were thawed and resuspended in buffer A containing 20 mM 3-(N-morpholino)propane-sulfonic acid (MOPS), pH 7.4, 100 mM NaCl, and then lysed by passing through a French Press at 1200 bar three times. Cell debris and non-lysed cells were removed by centrifugation at 14,000 rpm for 10 min at 4°C. The supernatant was collected and ultra-centrifuged at 36,900 rpm and 4°C for 2 hr. The membrane fraction was solubilized by addition of 1% (w/v) lauryl maltose neopentyl glycol (LMNG) in buffer A and incubated for 2 hr at 4°C with slow stirring. The suspension was ultra-centrifuged, and the supernatant was applied to Ni-NTA agarose beads (GE Healthcare) at 4°C. The beads were further washed in buffer A with 50 mM imidazole and 0.004% (w/v) LMNG. The buffer was exchanged to buffer B (20 mM MOPS, pH 7.4, 100 mM NaCl, and 0.1% [w/v] digitonin) and then washed in resin in batch mode. The protein was eluted from the beads with buffer B containing 500 mM imidazole. Protein was then concentrated and loaded onto a Superdex 6 increase (10/300GL, GE Healthcare) column equilibrated in buffer B. Peak fractions were pooled and concentrated to ~8 mg/mL for electron microscopy studies. The protein sample was analyzed by sodium dodecyl sulfate polyacrylamide gel electrophoresis (SDS-PAGE), and the bands were then identified through mass spectrometry.

## Activity assays

2,3-Dimethyl-1,4-naphthoquinone (DMNQ, CAS 2197-57-1) was synthesized by WuXi AppTec. In order to obtained reduced DMNQH$_2$, 20 mM DMNQ was ultrasonically dissolved in 1 mL ethanol with 6 mM HCl and reduced with sodium borohydride (NaBH$_4$) in the ice bath. An appropriate amount of 12 N HCl was added into quench unreacted NaBH$_4$ under the protection of argon. Oxidase activity was determined by a method described previously (*Bonner et al., 1986*; *Kusumoto et al., 2000*; *Safarian et al., 2019*). The oxygen consumption was monitored using the Clark-type oxygen electrode (Oxytherm[+], Hansatech) in 1 mL reaction buffer (20 mM MOPS, pH 7.4, 100 mM NaCl, 0.004% LMNG) at 25°C, containing 62.5 nM hybrid supercomplex. The reaction was started by addition of 100 μM DMNQH$_2$. The oxygen consumption curve was plotted using GraphPad Prime 8.0 software, from which an estimate of the oxygen-reduction rate ($v_0$) was obtained (corrected for autoxidation). In the inhibition assay, 500 nM Q203 or TB47, which is approximately the value of the median inhibitory concentration (IC50) according to our previous study (*Gong et al., 2018*), was chosen and incubated with 62.5 nM hybrid supercomplex for 20 min at 25°C. Inhibition curves and oxygen-reduction rates ($v_i$) with the different inhibitors were recorded. The inhibition rate (1 $v_0/v_i$) of Q203 and TB47 at 500 nM is reported. This assay was conducted using four groups of parallel experiments.

## Cryo sample preparation and data collection

300-mesh Quantifoil R0.6/1.0 grids (Quantifoil, Micro Tools GmbH, Germany) were glow-discharged at H$_2$/O$_2$ atmosphere for 25 s. 3 μL aliquots of protein complex at a concentration of 10 mg/mL were applied to the grid and then blotted for 3 s with force 0 at 8°C and 100% humidity using a Vitrobot IV (Thermo). Images were collected using a Titan Krios 300 keV electron microscope (Thermo), equipped with K3 Summit direct electron detector camera (Gatan). Data were recorded at 29,000× magnification with a calibrated super-resolution pixel size 0.82 Å/pixel. The exposure time was set to 2.4 s with 40 subframes and a total dose of 60 electrons per Å[2]. All images were automatically recorded using SerialEM with a defocus range from 1.2 μm to 1.8 μm (*Mastronarde, 2003*). For the datasets of apo, Q203-bound and TB47-bound *M. tuberculosis* cytochrome *bcc*, a total of 4141, 3763, and 2968 images were collected, respectively.

## Image processing

All dose-fractioned stacks were motion-corrected and dose-weighted using MotionCorr2 (*Zheng et al., 2017*) in RELION 3.03 (*Zivanov et al., 2019*). CTF estimation was conducted using cryoSPARC patch CTF estimation (*Punjani et al., 2017*). For the dataset of apo hybrid *M. tuberculosis* cytochrome *bcc*, 1,208,054 particles were picked automatically using EMD-9610 map as the template and extracted with a box size of rescaled 256 pixels (binned 2). 327,188 particles were selected after two rounds of 2D classification. 100,000 particles were used to perform *ab initio* reconstruction in four classes, and these four classes were used as 3D volume templates for heterogeneous refinement with all selected particles. 112,804 particles converged into one class with clear signals and then re-extracted with 512 pixels (binned 1). Next, this particle set was used to do homogeneous refinement and local refinement, yielding the final resolution 2.68 Å. For the dataset of Q203-bound and TB47-bound *M. tuberculosis* cytochrome *bcc*, the data processing was performed in a similar pipeline, resulting in the final reconstruction resolution at 2.67 Å and 2.93 Å, respectively (detailed parameters shown in supplementary figures).

## Model building and validation

The *M. smegmatis* respiratory complex $CIII_2CIV_2$ (PDB code: 6ADQ) model (*Gong et al., 2018*) as rigid body was fitted into EM density maps using UCSF Chimera 1.12 (*Pettersen et al., 2004*). Next, the resultant atomic model was manually modified according to the subunit sequences of *M. tuberculosis* cytochrome *bcc* and refined in COOT 0.8.9.1 (*Emsley et al., 2010*), followed by real-space refinement in PHENIX (*Adams et al., 2010*). The smile strings of Q203 and TB47 were generated and copied from ChemDraw (*Li et al., 2004*) and defined in PHENIX elBOW. Q203 and TB47 were manually built into the corresponding EM densities. The local resolution map was calculated in cryoSPARC (*Punjani et al., 2017*). All reported resolutions were based on the gold-standard FSC 0.143 criteria (*Rosenthal and Henderson, 2003*).

## MK9 docking study

The binding pose of MK9 with QcrB was generated by molecular docking using the Glide (Schrödinger, LLC). The spatially neighboring subunits QcrA/B were extracted from the hybrid supercomplex and used as the model. The protein structure was processed using the Protein Preparation Wizard module. In this process, hydrogens were added to heavy atoms and bond orders were assigned to each bond. The protonation state of each amino acid was predicted at pH 7.0 using the Epik algorithm (*Shelley et al., 2007*). All resolved waters were removed from the structure. At the same time, the structure of MK9 was prepared using the LigPrep module, and a low-energy conformation of MK9 was generated. The binding pocket of Q203 in QcrB was defined as the docking site using the Receptor Grid Generation module. The docking site was confined to an enclosed cube with side length of 20 Å, which was centered on the centroid of Q203. MK9 was then docked using the Ligand Docking module. Finally, 10 docking poses were generated for MK9 after post-docking minimization based on the force field of OPLS3 (*Harder et al., 2016*). These poses were ranked by the scoring function of GlideScore. The pose with the lowest GlideScore was chosen for binding mode analysis.

## Binding free energies of Q203 between wild-type and mutant hybrid supercomplex

Binding free energies of Q203 between wild-type and mutant hybrid supercomplex were computed using the Molecular Mechanics/Poisson-Boltzmann Surface Area (MM/PBSA) method based on all-atom molecular dynamic simulations (*Greene et al., 2019*; *Homeyer and Gohlke, 2012*; *Homeyer and Gohlke, 2015*). This task can be divided into three phases: (i) simulation system construction, (ii) molecular dynamic simulation, and (iii) binding free energy computation.

Considering that QcrB is the core subunit interacting with Q203 in the hybrid supercomplex, the QcrB with Q203 bound was the only region extracted from the Q203-bound hybrid supercomplex for simulation system construction. Membrane Builder in CHARMM-GUI (*Jo et al., 2007*) was used to build the protein/membrane system solvated in water. PDB file containing the coordination information of Q203-bound QcrB was read while only protein residues and Q203 were retained. The CHARMM General Force Field (CGenFF) was applied to parameterize the Q203 (*Vanommeslaeghe and MacKerell, 2012*). DPPC lipid molecules were used to explicitly build the lipid bilayers and bulky

water layers were added to the top and bottom sides of the membrane. In addition, 0.15 M KCl was added to neutralize the charge of system using the distance-based ion placing method. Next, the protein was inserted in the lipid bilayers and covered by the water layers despite exposure of some residues to the solvents. In order to generate topology and parameter files for subsequent AMBER dynamic simulations, the force fields of ff19SB (*Tian et al., 2020*) and Lipids17 (*Lee et al., 2020a*) were respectively applied to the parameterization of protein and lipid molecules. Four Q203-bound QcrB simulation systems were built, including wild-type QcrB, T313A mutant, E314Y mutant, and T313A/E314Y mutant.

Molecular dynamic simulation was carried out using AMBER 2020 (*Case et al., 2005*). Before dynamic simulation, 5000 steps of energy minimization were performed on waters to remove potential steric clashes between the solute and solvents. Harmonic restraints with a force constant of 10 kcal/mol $Å^2$ and 2.5 kcal/mol $Å^2$ were placed on protein and lipids in the minimization, respectively. Then, two stages of equilibrium simulations were carried out to relax the protein and lipid molecules by gradually decreasing the restraint force. The first stage was further divided into three short simulations each taking 375 ps with the time step set to 1 fs. The constant volume and temperature (NVT) ensemble was applied to this stage of simulation. The second one was a much longer simulation with the constant pressure and temperature (NPT) ensemble, which took 1.5 ns to completely relax the whole system with extremely low force. At this stage, the time step was increased to 2 fs while the SHAKE algorithm was used to fix the bonds with hydrogens (*Bailey and Lowe, 2009*). Subsequently, a 10 ns NPT simulation without any restraints was performed to further relax the system. Langevin dynamics was used to control the temperature at 303.15 K in NVT and NPT simulations (*Grønbech-Jensen and Farago, 2014*). In the NPT simulations, additional semi-isotropic pressure coupling and constant surface tension were applied to the simulation of lipid bilayers (*Bennun et al., 2007*). Once the equilibrium simulation was finished, a 100 ns NPT simulation free of restraints was carried out for each molecular system to produce the final trajectory used for energy calculation. Coordinates were printed every 100 ps so that in total 1000 frames were contained in the final trajectory. The CPPTRAJ module in AMBER was applied to analyze the trajectory (*Roe and Cheatham, 2013*).

MM/PBSA methods were employed to calculate the binding free energy of Q203 in wild-type and mutant QcrB (*Homeyer and Gohlke, 2012*). As shown in the equation ($\Delta G = E_{MM} + E_{pol} + E_{np} - TS$), free energy can be calculated by combining the MM part ($E_{MM}$) that represents the gas phase energy contribution as well as the solvation free energy components including polar ($E_{pol}$) and non-polar ($E_{np}$) contributions. The gas phase free energy was calculated according to the force field. As for the solvation free energy, the polar part representing electrostatic contribution was calculated using the Poisson–Boltzmann (PB) equation based on implicit solvent model (*Honig and Nicholls, 1995*). In AMBER, the equation also provides an implicit membrane model to calculate the membrane-mediated electrostatic interactions. Here, geometric multigrid based on iterative solver was selected for PB equation calculation (*Harris et al., 2013*). On the other hand, the LCPO method was applied to calculate the non-polar part that represents the hydrophobic contribution (*Weiser et al., 1999*). The Python script MMPBSA.py (*Miller et al., 2012*) in AMBER was used for the computation based on previously produced simulation trajectories, from which one frame was extracted every 1 ns for the energy calculations. Ions, water, and lipid molecules were all striped from the molecular systems before calculation. For implicit membrane model, the membrane dielectric constant was set to 7.0, and membrane thickness was set to 42 Å. Entropy contributions (TS) could be neglected as the four molecular systems were assumed to have similar entropy changes due to the same protein-ligand structures except for the mutated residues.

## Creation of figures

All the figures were created using UCSF Chimera (*Pettersen et al., 2004*) or PyMOL (*Schrödinger LLC, 2017*).

## Acknowledgements

We thank Dr. Chao Peng of the Mass Spectrometry System at the National Facility for Protein Science in Shanghai (NFPS), Zhangjiang Lab, SARI, China, for data collection and analysis and Prof. Lei Li (College of Life Sciences, Nankai University) for their technical support on Clark-type oxygen electrode and oxygen consumption assay. We thank Prof. Kaixia Mi (CAS Key Laboratory of Pathogenic

Microbiology and Immunology, Institute of Microbiology, CAS) for sharing the strain *M. smegmatis* mc$^2$ 51. We would like to thank Prof. Gregory M Cook (School of Biomedical Sciences, University of Otago, New Zealand) and Prof. Xiaoyun Lu (School of Pharmacy, Ji'nan University, China) for kindly providing TB47 for this study. We would also like to thank the Bio-Electron Microscopy Facility of ShanghaiTech University.

## Additional information

### Funding

| Funder | Grant reference number | Author |
|---|---|---|
| National Key Research and Development Program of China | 2017YFC0840300 | Zihe Rao |
| National Key Research and Development Program | 2020YFA0707500 | Zihe Rao |
| National Natural Science Foundation of China | 81520108019 | Zihe Rao |
| National Natural Science Foundation of China | 813300237 | Zihe Rao |
| National Natural Science Foundation of China | 32100976 | Hongri Gong |
| Natural Science Foundation of Tianjin City | 20JCQNJC01430 | Hongri Gong |

The funders had no role in study design, data collection and interpretation, or the decision to submit the work for publication.

### Author contributions

Shan Zhou, Data curation, Formal analysis, Methodology, Writing – original draft, Writing – review and editing; Weiwei Wang, Data curation, Formal analysis, Methodology, Visualization, Writing – review and editing; Xiaoting Zhou, Yuying Zhang, Yuezheng Lai, Yanting Tang, Jinxu Xu, Dongmei Li, Jianping Lin, Xiaolin Yang, Quan Wang, Formal analysis, Methodology; Ting Ran, Hongming Chen, Data curation, Formal analysis, Methodology; Luke W Guddat, Formal analysis, Methodology, Writing – review and editing; Yan Gao, Data curation, Formal analysis, Methodology, Writing – review and editing; Zihe Rao, Funding acquisition, Resources, Supervision, Writing – review and editing; Hongri Gong, Conceptualization, Formal analysis, Funding acquisition, Investigation, Methodology, Supervision, Visualization, Writing – original draft, Writing – review and editing

### Author ORCIDs

Shan Zhou http://orcid.org/0000-0001-5468-3538
Xiaoting Zhou http://orcid.org/0000-0002-8238-8242
Yanting Tang http://orcid.org/0000-0002-8656-3220
Xiaolin Yang http://orcid.org/0000-0003-0992-8676
Ting Ran http://orcid.org/0000-0002-1387-4634
Hongming Chen http://orcid.org/0000-0002-8065-8333
Luke W Guddat http://orcid.org/0000-0002-8204-8408
Yan Gao http://orcid.org/0000-0002-9665-8645
Zihe Rao http://orcid.org/0000-0001-9866-2384
Hongri Gong http://orcid.org/0000-0002-2596-7635

### Decision letter and Author response

Decision letter https://doi.org/10.7554/eLife.69418.sa1
Author response https://doi.org/10.7554/eLife.69418.sa2

# Additional files

## Supplementary files
• Transparent reporting form

## Data availability
All data generated or analysed during this study are included in the manuscript and supporting files. Source data files have been provided. The accession numbers for the 3D cryo-EM density map of apo, Q203-bound and TB47-bound hybrid supercomplex in present study are EMD-30943, EMD-30944 and EMD-30945, respectively. The accession numbers for the coordinates for the apo, Q203-bound and TB47-bound hybrid supercomplex in this study are PDB: 7E1V, PDB: 7E1W and PDB: 7E1X, respectively.

The following dataset was generated:

| Author(s) | Year | Dataset title | Dataset URL | Database and Identifier |
|---|---|---|---|---|
| Zhou S, Wang W, Gao Y, Gong H, Rao Z | 2021 | Cryo-EM structure of apo hybrid respiratory supercomplex consisting of Mycobacterium tuberculosis complexIII and Mycobacterium smegmatis complexIV | https://www.ebi.ac.uk/emdb/EMD-30943 | Electron Microscopy Data Bank, EMD-30943 |
| Zhou S, Wang W, Gao Y, Gong H, Rao Z | 2021 | Cryo-EM structure of hybrid respiratory supercomplex consisting of Mycobacterium tuberculosis complexIII and Mycobacterium smegmatis complexIV in the presence of Q203 | https://www.ebi.ac.uk/emdb/EMD-30944 | Electron Microscopy Data Bank, EMD-30944 |
| Zhou S, Wang W, Gao Y, Gong H, Rao Z | 2021 | Cryo-EM structure of hybrid respiratory supercomplex consisting of Mycobacterium tuberculosis complexIII and Mycobacterium smegmatis complexIV in presence of TB47 | https://www.ebi.ac.uk/emdb/EMD-30945 | Electron Microscopy Data Bank, EMD-30945 |
| Zhou S, Wang W, Gao Y, Gong H, Rao Z | 2021 | Cryo-EM structure of apo hybrid respiratory supercomplex consisting of Mycobacterium tuberculosis complexIII and Mycobacterium smegmatis complexIV | https://www.rcsb.org/structure/7E1V | RCSB Protein Data Bank, 7E1V |
| Zhou S, Wang W, Gao Y, Gong H, Rao Z | 2021 | Cryo-EM structure of hybrid respiratory supercomplex consisting of Mycobacterium tuberculosis complexIII and Mycobacterium smegmatis complexIV in the presence of Q203 | https://www.rcsb.org/structure/7E1W | RCSB Protein Data Bank, 7E1W |
| Zhou S, Wang W, Gao Y, Gong H, Rao Z | 2021 | Cryo-EM structure of hybrid respiratory supercomplex consisting of Mycobacterium tuberculosis complexIII and Mycobacterium smegmatis complexIV in presence of TB47 | https://www.rcsb.org/structure/7E1X | RCSB Protein Data Bank, 7E1X |

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
