## [Editor Report]

The authors describe the cryo EM structures of cytochrome bcc from mycobacteria in complex with clinical drug candidates Q203 and TB47, which are being trialled for the treatment of tuberculosis. The cryo EM structures were obtained by expressing and purifying a hybrid super complex; the bcc part from *M. tuberculosis* and the cytochrome oxidase part from *M. smegmatis*. The cryo EM structures show how the Q203 and TB47 inhibitors bind and block the binding of quinone at the Qo site, thus inhibiting electron-transfer. The complex structures should facilitate the structure-based development of Q203 and TB47 drug compounds against tuberculosis.

---

## [Decision Letter]

**Decision letter after peer review:**

Thank you for submitting your article "Structure of mycobacterial cytochrome bcc in complex with Q203 and TB47, two anti-TB drug candidates" for consideration by *eLife*. Your article has been reviewed by 3 peer reviewers, including David Drew as the Reviewing Editor and Reviewer #1, and the evaluation has been overseen by Michael Marletta as the Senior Editor.

Essential revisions:

The main impact of this paper is the drug interaction. We think it is essential that both a mechanism of inhibition and selectivity for the Q203/TB47 drugs are addressed, i.e, an atomic model on its own is insufficient to reach these conclusions.

1) The authors need to demonstrate activity of the chimeric supercomplex (from oxygen consumption experiments or any other relevant activity assay) and validate that Q203 and TB47 inhibit as expected.

2) The affinity of Q203 (i.e., with ITC) in combination with mutagenesis and or computational approaches are required to elucidate the molecular basis how the selectivity of the drug compound to *M. tuberculous* is accomplished.

3) We require a deeper discussion regarding the super complex of the hybrid and differences in complex III in *M. tuberculous* compared to *M. smegmatis* to evaluate if the structure yields any new mechanistic insights.

*Reviewer #1 Recommendations for the Authors:*

1. A rigorous comparison, using mutagenesis and binding and functional assays and/or computational methods is essential to provide a meaningful model for establishing how drug specificity for *M. tuberculosis* cytochrome bcc is achieved by these compounds.

2. An MK molecule was identified, but it was located some 15 Å from the heme bL and is thought that the actual binding site for MK would be where the drugs are binding. What is the relevance for the bound MK molecule? Why was it not possible to model menaquinol in the proposed location and has this quinone site been structurally shown previously for other bcc complexes? Do the authors have a reason for this and why do quinone binding sites show species-specific differences? Why is it not possible to directly measure competition between the drugs and menaquinol?

3. The authors have not elaborated on any structural differences of previous cytochrome bcc complexes and how this might br relevant for understanding electron-driven proton transport in *M. tuberculosis*. Indeed, the Mtb cytochrome bcc is complex with cytochrome c oxidase from a different bacterial host *M. smegmatis*. What was the practical reason for expressing as a hybrid complex? How does the hybrid complex compare to the supercomplex from *M. smegmatis*? The paper would clearly benefit from more functional comparisons.

*Reviewer #2 Recommendations for the Authors:*

I find this work interesting, but certain shortcomings should be addressed before I can make a final recommendation.

1. Very little is said about the complex IV part of the resolved supercomplex. The authors should clearly explain how the hybrid supercomplex is affecting the structure, including possible artifacts arising from non-native contacts.

2. What is the activity of this chimeric supercomplex? What is the binding affinity of Q203 and TB47 and do they actually inhibit the activity?

*Reviewer #3 Recommendations for the Authors:*

1. Was the purified enzyme active? Please provide data from oxygen consumption experiments or any other relevant activity assay.

2.. To which degree was the purified enzyme susceptible to inhibition by Q203 and TB47?

3.. Lines 172 – 190 and Figure 6: the comparison of bcc/bc1 structures provides information regarding why Q203 does not inhibit homologies complexes from non-mycobacterial species. What do these structure data imply for *M. smegmatis*? This structure is shown in Figure 6 without steric clashes indicated, but is not referred to in main text. How do the structural data regarding Q203 binding to *M. smegmatis* bcc correlate to the sensitivity of this model strain for Q203?

[Editors' note: further revisions were suggested prior to acceptance, as described below.]

Thank you for resubmitting your work entitled "Structure of mycobacterial cytochrome bcc in complex with Q203 and TB47, two anti-TB drug candidates" for further consideration by *eLife*. Your revised article has been reviewed by 2 peer reviewers and the evaluation has been overseen by Cynthia Wolberger as the Senior Editor and a Reviewing Editor.

The manuscript has been improved but there are a few remaining issues that need to be addressed, as outlined below.

Summary:

The authors describe the cryo EM structures of cytochrome bcc from mycobacteria in complex with clinical drug candidates Q203 and TB47, which are being trialled for the treatment of tuberculosis. The cryo EM structures were obtained by expressing and purifying a hybrid super complex; the bcc part from *M. tuberculosis* and the cytochrome oxidase part from *M. smegmatis*. The cryo EM structures show how the Q203 and TB47 inhibitors bind and block the binding of quinone at the Qo site, thus inhibiting electron-transfer. The complex structures should facilitate the structure-based development of Q203 and TB47 drug compounds against tuberculosis.

1. It would be helpful to give the timescale in the SI figure 6-suppl 2 rather than a frame number, which is cryptic for the general reader. The authors should also better describe what the different traces mean, for example what is the implication of the high RMSD for R313A+E314Y in panel B?

2. The binding free energies presented in Table 4 seem unphysically large (around -65 kcal/mol, which would result in binding constant of 10^-^(65/RT)), which could be misleading. Reporting relative values might be more appropriate here.

*Reviewer #2 (Recommendations for the authors):*

The authors have adequately addressed most of my comments, and this has resulted in a significantly improved manuscript. Importantly, the authors provide in the revised manuscript activity measurements, which show that the WT complex is active, and that the inhibitors significantly reduce the activity. The addition of missing residues in the figures, and explicitly showing the densities for selected residues and ligands have also improved the discussion. The authors could add a comment on what the biological function of binding so many MK molecules to the supercomplex might be. Although no experimental estimates of the binding affinities could be provided, the authors now present some computational simulations to estimate the binding effects. In this regard, it would be helpful to give the timescale in the SI figure 6-suppl 2 rather than a frame number, which is cryptic for the general reader. The authors should also better describe what the different traces mean, for example what is the implication of the high RMSD for R313A+E314Y in panel B. Also, the binding free energies presented in Table 4 seem unphysically large (around -65 kcal/mol, which would result in binding constant of 10^-(65/RT)!), which could be misleading. As far as I understand, the method estimates only relative binding affinities based on an approximate method. Reporting relative values might be more appropriate.

*Reviewer #3 (Recommendations for the authors):*

The authors have addressed my questions. I recommend accepting this paper.

---

## [Author Response]

Essential revisions:The main impact of this paper is the drug interaction. We think it is essential that both a mechanism of inhibition and selectivity for the Q203/TB47 drugs are addressed, i.e, an atomic model on its own is insufficient to reach these conclusions.1) The authors need to demonstrate activity of the chimeric supercomplex (from oxygen consumption experiments or any other relevant activity assay) and validate that Q203 and TB47 inhibit as expected.

We have now shown that the purified chimeric supercomplex is a functional assembly with a turnover number of 23.3 +/- 2.4 e^-^s^-1^ (mean ± s.d., n = 4), in agreement with the previous study that shows *M. tuberculosis* CIII can functionally complement native *M. smegmatis* CIII and maintain the growth of *M. smegmatis* (Kim et al., 2015). The in vitro inhibitions of this enzyme by Q203 and TB47 was determined by means of an DMNQH_2_/oxygen oxidoreductase activity assay. In the assay, 500 nM Q203 or TB47 was chosen, which is close to the median inhibitory concentration (IC50) obtained from the menadiol-induced oxygen consumption in our previous study (Gong et al., 2018). After addition of Q203 and TB47, the values of turnover number of the hybrid supercomplex are reduced to 5.8 +/- 2.4 e^-^s^-1^ (Figure 4—figure supplement 4) and 5.1 +/- 2.9 e^-^s^-1^ (Figure 5—figure supplement 4) respectively, from 23.3 +/- 2.4 e^-^s^-1^. We have incorporated this new data into the text (lines 90-93, 187-189, 206-209).

2) The affinity of Q203 (i.e., with ITC) in combination with mutagenesis and or computational approaches are required to elucidate the molecular basis how the selectivity of the drug compound to M. tuberculous is accomplished.

It is not possible to detect the binding of Q203 and TB47 using isothermal titration calorimetry (ITC) or microscale thermophoresis (MST). This is due to the complicated identity of the multiple subunits of membrane protein complex, the limited solubility of the drugs and the limited yield of the hybrid supercomplex. In addition, the purified complex is bound to endogenous substrate at the drug-binding site, which also indirectly affects the accurate measurement of the affinity of the drugs. Therefore, an alternative approach is needed to measure the true binding affinity of the Q203 in the future.

Here we have been able to calculate the binding free energies of Q203 between wild-type and mutant hybrid supercomplex. This data has been used to further elucidate the importance of some key residues that could play a role in determining the inhibitor specificity. According to a previous functional study (Pethe et al., 2013), structural comparison (Figure 6) and sequence alignment (Figure 6—figure supplement 1), _QcrB_Thr^313^ and _QcrB_Glu^314^ of hybrid supercomplex were selected and mutated to their counterparts in the human supercomplex, namely residues Ala and Tyr, respectively. It is believed that Ala would decrease the affinity of Q203 (Figure 4) (Pethe et al., 2013) and the Tyr could lead to steric hindrance of Q203 binding (Figure 6). 75 ns simulations were used for the calculation of binding free energy (Figure 6—figure supplement 2). The 75 ns simulation was further equally divided into three phases and the average values were used as the final binding free energy (Table 4). It is suggested that the wild-type system has stronger affinity than the mutant systems (Thr313Ala or Glu314Tyr). The free energy of binding calculation for Q203 in the dual mutant (Thr313Ala and Glu314Tyr) indicates that its binding affinity is significantly weaker compared to the wild-type or single-mutant systems. Therefore, the suggestions drawn from the calculation of binding free energy of Q203 provide further evidence for specificity of Q203 inhibition. We have incorporated discussion into the text (line 234-248).

3) We require a deeper discussion regarding the super complex of the hybrid and differences in complex III in M. tuberculous compared to M. smegmatis to evaluate if the structure yields any new mechanistic insights.

The hybrid supercomplex including the *M. tuberculosis* CIII dimer forms a pseudo 2-fold symmetrical compact rod, but with a slight curvature in the membrane plane, as previously observed in the *M. smegmatis* CIII_2_CIV_2_ supercomplex (Gong et al., 2018; Wiseman et al., 2018) (Figure 2A, B). The hybrid supercomplex is structurally similar to the supercomplex isolated from *M. smegmatis* (Gong et al., 2018; Wiseman et al., 2018) (Figure 2—figure supplement 4), except that subunits LpqE and Unk. (probably MSMEG_0987) (Wiseman et al., 2018) were not observed here. The absence of these two subunits may be due to their depletion during purification of the supercomplex or their map density signal was averaged to background noise during structural determination. As expected, the topology of *M. tuberculosis* cytochrome *bcc* and *M. smegmatis* CIV in the hybrid supercomplex is also similar to that of the equivalent subunits in the *M. smegmatis* CIII_2_CIV_2_ supercomplex (Figure 2C, D; Figure 2-figures supplement 4, 5). As such, there is no notable non-native contacts that resulted from the hybrid assembly in the hybrid supercomplex. This is attributable to the high structural similarity between the *M. tuberculosis* cytochrome *bcc* and *M. smegmatis* cytochrome *bcc.* In the *M. smegmatis* CIII_2_CIV_2_ supercomplex, the cytochrome *cc* head domain of QcrC adopts an ‘open’ or a ‘closed’ conformation (Gong et al., 2018; Wiseman et al., 2018). However, in this structure it is only the closed conformation (Figure 2—figure supplement 4). Considering the high topology similarity of the subunits and the arrangement of prosthetic groups compared to previous *M. smegmatis* CIII_2_CIV_2_ supercomplex, the mechanism of action of the hybrid supercomplex including *M. tuberculosis* cytochrome *bcc* is expected to be the same as that for the *M. smegmatis* CIII_2_CIV_2_ supercomplex (Gong et al., 2018). These new discussion points have been incorporated into the revised manuscript (lines 96-108, 120-127).

Reviewer #1 Recommendations for the Authors:1. A rigorous comparison, using mutagenesis and binding and functional assays and/or computational methods is essential to provide a meaningful model for establishing how drug specificity for M. tuberculosis cytochrome bcc is achieved by these compounds.

See answer to Question 2 from the major Essential Revisions.

2. An MK molecule was identified, but it was located some 15 Å from the heme bL and is thought that the actual binding site for MK would be where the drugs are binding. What is the relevance for the bound MK molecule? Why was it not possible to model menaquinol in the proposed location and has this quinone site been structurally shown previously for other bcc complexes? Do the authors have a reason for this and why do quinone binding sites show species-specific differences? Why is it not possible to directly measure competition between the drugs and menaquinol?

We have now rephrased the corresponding sentences and discussed in depth. The edge-to-edge distance from MK to heme *b*_L_ is 15 Å, exceeding the 14 Å limit for efficient physiological electron transfer (Page et al., 1999). Furthermore, there are no observed hydrogen bonds to the carbonyl groups of MK, which are needed to help deprotonate MKH_2_. In addition, structural superposition with the inhibitor-bound *bc*_1_ complex also shows that the MK binds deeper into the Q_P_ pocket in the *bc*_1_ complex (Birth et al., 2014; Solmaz and Hunte, 2008) than in the currently described complex (Figure 3—figure supplement 2). Hence, the endogenous electron donor MKH_2_ should bind deeper inside the pocket and close to polar residues such as _QcrB_Tyr^161^, _QcrB_Thr^313^, and _QcrB_Asp^314^ (Figure 3D), to facilitate electron transfer. In addition, MK could also be modeled to fit deeper inside the pocket (Figure 3—figure supplement 3). Thus, we speculate that what is observed here is the oxidized product as it leaves the Q_P_ site.

Quinone-binding sites are essential to the function of respiratory chain complexes and thus are good targets for drug discovery (Harikishore et al., 2020; Lee et al., 2020a). Structurally diverse quinones such as ubiquinone and menaquinone bind in the mitochondrial ETC (Zhang et al., 1998) and mycobacterial ETC (Gong et al., 2018; Wiseman et al., 2018), respectively, indicating there are structural differences in the quinone-binding sites of the different species. Structural differences in the ubiquinone-binding site are also observed when the mitochondrial and *Escherichia coli* respiratory chain complex IIs are compared (Huang et al., 2021). Therefore, it is feasible that species-specific quinone-binding-site inhibitors could be designed. We have incorporated above discussions into the revised text (lines 129-136, 147-156).

It is not possible to detect the binding of the drugs (Q203 and TB47) and menaquinol (native substrate, MK9) using isothermal titration calorimetry (ITC) or microscale thermophoresis (MST). This is due to the complicated identity of the multiple subunits of membrane protein complex, the limited solubility of the drugs and the limited yield of the hybrid supercomplex. In addition, the purified complex is bound to endogenous substrate at the drug-binding site, which also indirectly affects the accurate measurement of the affinity of the drugs and menaquinol. An alternative approach is needed to measure the true binding affinity of the drugs and menaquinol in the future.

3. The authors have not elaborated on any structural differences of previous cytochrome bcc complexes and how this might br relevant for understanding electron-driven proton transport in M. tuberculosis. Indeed, the Mtb cytochrome bcc is complex with cytochrome c oxidase from a different bacterial host M. smegmatis. What was the practical reason for expressing as a hybrid complex? How does the hybrid complex compare to the supercomplex from M. smegmatis? The paper would clearly benefit from more functional comparisons.

Information on the structural features and mechanism of action of hybrid complex including *M. tuberculosis* cytochrome *bcc* has been discussed above (See answer to Question 3 from the major Essential Revisions).

The reasons for expressing a hybrid complex are as follows. On the one hand, it is difficult to successfully express multiple subunits of a protein complex like mycobacterial CIII_2_CIV_2_ respiratory chain supercomplex, especially when the genes coding subunits of the complex are not clustered together. On the other hand, considering that the hybrid supercomplex has been confirmed as a stable and functional assembly and its acquisition can be operated practically (Kim et al., 2015), we therefore expressed and purified the hybrid supercomplex, and finally determined its structure.

Reviewer #2 Recommendations for the Authors:I find this work interesting, but certain shortcomings should be addressed before I can make a final recommendation.1. Very little is said about the complex IV part of the resolved supercomplex. The authors should clearly explain how the hybrid supercomplex is affecting the structure, including possible artifacts arising from non-native contacts.

See answer to Question 3 from the major Essential Revisions.

2. What is the activity of this chimeric supercomplex? What is the binding affinity of Q203 and TB47 and do they actually inhibit the activity?

See answer to Questions 1 and 2 from the major Essential Revisions.

Reviewer #3 Recommendations for the Authors:1. Was the purified enzyme active? Please provide data from oxygen consumption experiments or any other relevant activity assay.

See answer to Question 1 from the major Essential Revisions.

2. To which degree was the purified enzyme susceptible to inhibition by Q203 and TB47?

See answer to Question 1 from the major Essential Revisions.

3. Lines 172 – 190 and Figure 6: the comparison of bcc/bc1 structures provides information regarding why Q203 does not inhibit homologies complexes from non-mycobacterial species. What do these structure data imply for M. smegmatis? This structure is shown in Figure 6 without steric clashes indicated, but is not referred to in main text. How do the structural data regarding Q203 binding to M. smegmatis bcc correlate to the sensitivity of this model strain for Q203?

It is worth noting that the structures of cytochrome *bcc* from *M. tuberculosis* and *M. smegmatis* have high similarity (Figure 2-figures supplement 5), and no steric hindrance is observed between the Q203 and *M. smegmatis* cytochrome *bcc* (Figure 6). This observation indicates that Q203 should have a similar binding mechanism and a similar effect on the activity of cytochrome *bcc* from *M. smegmatis* and *M. tuberculosis*. This is in good agreement with previous antimycobacterial activity data and inhibition data for the *bcc* complexes from *M. smegmatis* and *M. tuberculosis* (Gong et al., 2018; Lu et al., 2018a). This discussion has now been added into the text (line 220-227).

[Editors' note: further revisions were suggested prior to acceptance, as described below.]

The manuscript has been improved but there are a few remaining issues that need to be addressed, as outlined below.Summary:The authors describe the cryo EM structures of cytochrome bcc from mycobacteria in complex with clinical drug candidates Q203 and TB47, which are being trialled for the treatment of tuberculosis. The cryo EM structures were obtained by expressing and purifying a hybrid super complex; the bcc part from M. tuberculosis and the cytochrome oxidase part from M. smegmatis. The cryo EM structures show how the Q203 and TB47 inhibitors bind and block the binding of quinone at the Qo site, thus inhibiting electron-transfer. The complex structures should facilitate the structure-based development of Q203 and TB47 drug compounds against tuberculosis.1. It would be helpful to give the timescale in the SI figure 6-suppl 2 rather than a frame number, which is cryptic for the general reader. The authors should also better describe what the different traces mean, for example what is the implication of the high RMSD for R313A+E314Y in panel B?

We have provided a new SI figure 6-suppl 2 with the timescale as the X-axis. The different values for the root mean squared deviation (RMSD) suggest that three mutants might have the different effects on the structural conformation of QcrB, and the dual mutant (Thr313Ala and Glu314Tyr) induced significantly structural changes compared to the wild-type complex. New discussion points have been incorporated into the revised manuscript (line 244-247).

2. The binding free energies presented in Table 4 seem unphysically large (around -65 kcal/mol, which would result in binding constant of 10^-(65/RT)), which could be misleading. Reporting relative values might be more appropriate here.

We thank the reviewer for this suggestion. We have reported the relative value for each mutant compared to the wild-type in the updated Table 4.

Reviewer #2 (Recommendations for the authors):The authors have adequately addressed most of my comments, and this has resulted in a significantly improved manuscript. Importantly, the authors provide in the revised manuscript activity measurements, which show that the WT complex is active, and that the inhibitors significantly reduce the activity. The addition of missing residues in the figures, and explicitly showing the densities for selected residues and ligands have also improved the discussion.The authors could add a comment on what the biological function of binding so many MK molecules to the supercomplex might be.

These quinones are observed to be remote from the canonical quinone-binding sites and into the membrane space, suggesting they have a structural rather than functional role. We have incorporated this discussion into the text (line 140-141).

Although no experimental estimates of the binding affinities could be provided, the authors now present some computational simulations to estimate the binding effects. In this regard, it would be helpful to give the timescale in the SI figure 6-suppl 2 rather than a frame number, which is cryptic for the general reader. The authors should also better describe what the different traces mean, for example what is the implication of the high RMSD for R313A+E314Y in panel B.

See answer to Question 1 from the Summary.

Also, the binding free energies presented in Table 4 seem unphysically large (around -65 kcal/mol, which would result in binding constant of 10^-(65/RT)!), which could be misleading. As far as I understand, the method estimates only relative binding affinities based on an approximate method. Reporting relative values might be more appropriate.

See answer to Question 2 from the Summary.